# PHASE-DRIVEN DOMAIN GENERALIZABLE LEARNING FOR NONSTATIONARY TIME SERIES CLASSIFICATION

## ABSTRACT

Monitoring and recognizing patterns in continuous sensing data is crucial for many practical applications. These real-world time-series data are often *nonstationary*, characterized by varying statistical and spectral properties over time. This poses a significant challenge in developing learning models that can effectively generalize across different distributions. In this work, based on our observation that nonstationary statistics for time-series classification tasks are intrinsically linked to the phase information, we propose a time-series domain generalization framework, `PhASER`. It consists of three key elements: 1) *Hilbert transform-based phase augmentation* that diversifies non-stationarity while preserving discriminatory semantics, 2) *separate magnitude-phase encoding* by viewing time-varying magnitude and phase as independent modalities, and 3) *phase-residual feature broadcasting* by incorporating phase with a novel residual connection for inherent regularization to enhance distribution invariant learning. Extensive evaluation on 5 datasets from sleep-stage classification, human activity recognition, and gesture recognition against 13 state-of-the-art baseline methods demonstrate that `PhASER` consistently outperforms the best baselines by an average of 5% and up to 11% in some cases. Moreover, `PhASER`'s principles can also be applied broadly to boost the generalizability of existing time-series classification models.

## 1 INTRODUCTION

Time-series data play a ubiquitous and crucial role in numerous real-world applications, such as continuous monitoring for human activity recognition (Li et al., 2020), gesture identification (Ozdemir et al., 2020), sleep tracking (Kemp et al., 2000), and more. Continuous time series often exhibit *non-stationarity*, i.e., the statistical and spectral properties of the data evolve over time. Another inherent challenge is the distribution shift due to the underlying sensing properties or subject-specific attributes, commonly referred to as *domain shift*, which directly degrades the performance of time-series models in real-world applications. Thus, developing methods for more generalizable pattern recognition in nonstationary time series classification is crucial.

Most existing methods (Ragab et al., 2023a;b; He et al., 2023) tackle distribution shifts in time-series applications via domain adaptation, assuming accessible target domain samples. Yet, obtaining data from unseen distributions in advance is not always feasible. To overcome this challenge, a few works (Gagnon-Audet et al., 2022; Xu et al., 2022) applied standard domain generalization (DG) algorithms (Volpi et al., 2018; Sagawa et al., 2019; Parascandolo et al., 2020) to temporally-varying time-series data, but reported a significant performance gap when compared with visual data. Recent research on DG tailored for time series explores latent-domain characterization (Lu et al., 2023; Du et al., 2021), augmentation strategies (Iwana & Uchida, 2021; Li et al., 2021), preservation of non-stationarity dictionary (Liu et al., 2022; Kim et al., 2021c), and utilization of spectral characteristics of time series (He et al., 2023; Yang & Hong, 2022; Kim et al., 2021a). While successful in some cases, these methods have their limitations. Latent-domain characterization heavily relies on the hypotheses of latent domains, limiting its broader applicability. Augmentation strategies (shift, jittering, masking, etc.) for time series may not be universally applicable and can impair the task (Iwana & Uchida, 2021). For instance, in physiological signal analysis, morphological alterations from augmentations are harmful, and time-slicing is unsuitable for periodic signals. Advanced augmentation techniques like spectral perturbations (time-frequency warping, decomposition techniques, etc.) are usually heavily parametric (Wen et al., 2021) and application-specific. Other approaches specific to preserving

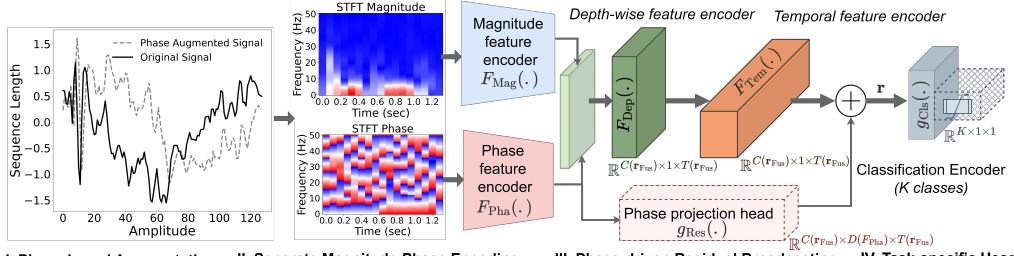

Figure 1: PhASER's components: I. Hilbert transform-based phase augmentation. II. Separate feature encoding of time-varying phase and magnitude derived from Short-Term Fourier Transform (STFT) using $F_{\mathrm{Mag}}$ and $F_{\mathrm{Pha}}$. III. Key elements of the phase-residual broadcasting network, demonstrating design of depth-wise feature encoder ($F_{\mathrm{Dep}}$), temporal encoder ($F_{\mathrm{Tem}}$), and incorporation of phase-projection head's output ($g_{\mathrm{Res}}$) for broadcasting (annotated dimensions of intermediate feature maps). IV. Task-specific classification encoder ($g_{\mathrm{Cls}}$).

non-stationarity are constrained by maintaining the same input-output space, making them unsuitable for multivariate time-series classification tasks. While some works (He et al., 2023; Yang & Hong, 2022) focus on frequency domain representations for robustness to feature shifts, they overlook cases with time-varying spectral responses. Another significant issue is that many of these studies rely on domain identity, which in practice is expensive and intrusive to obtain, especially in healthcare and finance (Yan et al., 2024; Bai et al., 2022). Thus, achieving domain-generalizable time-series classification without access to unseen distributions and domain labels of available distributions remains a challenging yet crucial pursuit.

**Our Approach and Contributions.** We propose a novel Phase-Augmented Separate Encoding and Residual (PhASER) framework to achieve domain-generalizable classification for *nonstationary* real-world time series. Figure 1 illustrates an overview of PhASER, which includes three key modules. First, we diversify the non-stationarity of source domain data through an intra-instance phase shift, by leveraging the generality and non-parametric nature of Hilbert Transform (HT) (King, 2009) to introduce a phase-shift-based augmentation. Next, we apply a novel strategy to encode the time-varying magnitude and phase responses separately for enhanced integration of the time-frequency information. Finally, we design an effective broadcasting mechanism with a non-linear residual connection between the phase-encoded embedding and the backbone representation to learn domain-invariant and generalizable (He et al., 2020; Marion et al., 2023) task-specific features (He et al., 2016). We experiment with 13 baselines on 5 datasets to quantitatively demonstrate PhASER's superiority in learning generalizable representations, even in challenging scenarios like transferring from one domain to multiple domains. Additionally, we provide design insights through ablation analysis, explore PhASER's applicability to other architectures, and present qualitative visualizations of its learned representations.

## 2 APPROACH

### 2.1 PROBLEM FORMULATION

**Definition 2.1** (**Nonstationary Time Series**). Following the definition of mixed decomposition-based nonstationary signals in Dama & Sinoquet (2021), we assume that a nonstationary time-series sample $\mathbf{x} = \{x_0, ..., x_t, ...\}$ drawn from a domain $\mathcal{D}_{\mathbf{x}}$ can be decomposed into components with mean $\mu_t$ and variance $\sigma_t$ (both $\mu_t$ and $\sigma_t$ are not always zero) as:

$$\mathrm{Pr}_{\mathbf{x} \sim \mathcal{D}_{\mathbf{x}}}(\mathbf{x})(t) = \mu_t + \sigma_t \times z, \text{ where } \forall L \geq 1, \exists t, [\mu_t \neq \mu_{t+L}] \vee [\sigma_t \neq \sigma_{t+L}], \tag{1}$$

where $z$ is a stationary stochastic component with a zero mean and a unit variance.

**Definition 2.2** (**Time-Series Domain Generalization**). Suppose there is a dataset $\mathbf{S} = \{(\mathbf{x}_i, y_i)\}_{i=1}^{M}$ with $M$ nonstationary time-series samples drawn from a set of $N_S$ source domains $S = \{\mathcal{S}_i\}_{i=1}^{N_S}$. The joint distribution of $\mathbf{S}$ is $\mathrm{Pr}(\mathcal{X}_{\mathbf{S}}, \mathcal{Y}_{\mathbf{S}})$, i.e., $\mathbf{x}_i \sim \mathcal{X}_{\mathbf{S}}, y_i \sim \mathcal{Y}_{\mathbf{S}}$ and $\mathbf{x}_i \in \mathbb{R}^{V \times T}$, where $V$ is the number of time-series feature dimensions and $T$ is the sequence length. $y_i \in \mathbb{R}^{1 \times 1}$ is the categorical

label. Note that the joint distributions of different source domains are similar (with shared underlying patterns) but domain-specific distinctions:

$$\Pr(\mathcal{X}_{\mathcal{S}_i}, \mathcal{Y}_{\mathcal{S}_i}) \neq \Pr(\mathcal{X}_{\mathcal{S}_j}, \mathcal{Y}_{\mathcal{S}_j}), 1 < i \neq j \leq N_S. \tag{2}$$

For any potential unseen target domain $\mathcal{D}_U$, its joint distribution remains distinct like Eq. (2). In our problem, although the source dataset is assumed to contain multiple domains, the annotations that specify the domain identity are unavailable. Our goal is to train a model consisting of a feature extractor $F$ and a classifier $g$ using the given source dataset ($F \circ g : \mathcal{X}_S \to \mathcal{Y}_S$), such that

$$\min \underset{(\mathbf{x},y) \sim \mathcal{D}_U}{\mathbb{E}} [\mathcal{L}(g(F(\mathbf{x})), y)], \tag{3}$$

where $\mathcal{L}(\cdot)$ is a certain cost that measures the errors between model predictions and the ground truth.

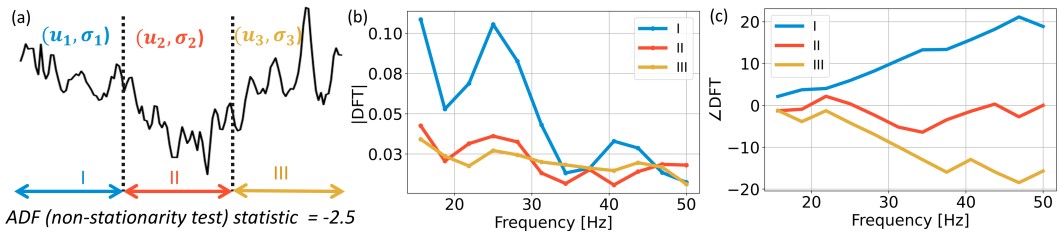

Figure 2: Illustrative example of non-stationarity using a sample from a human activity recognition dataset (HHAR) where (a) shows the temporal non-stationarity of a signal denoted by varying mean $\mu$ and variance $\sigma$ within a domain for three regions color-coded and denoted as I, II, and III. (b) shows that the magnitude response ($|\text{DFT}|$) of the Discrete Fourier Transform (DFT) for each region is distinct. There is a clear difference in the dominant frequency for each region. (c) shows the phase responses ($\angle(\text{DFT})$) for each region. The $\angle(\text{DFT})$ of each region is also distinct.

**Motivation.** We motivate our study through a human activity recognition (HAR) application, where non-stationarity is unavoidable due to changes in user behavior or sensor characteristics (Bangaru et al., 2020). We illustrate an instance of non-stationarity in Figure 2 (a), which visualizes a univariate accelerometer data sample from a dataset called HHAR (Stisen et al., 2015) in the time domain. By segmenting this sample into sequential windows and conducting a Discrete Fourier Transform (DFT) to obtain its magnitude and phase responses, as shown in Figures 2 (b) and (c), we observe the shifts in the spectral domain that correspond to non-stationarity.

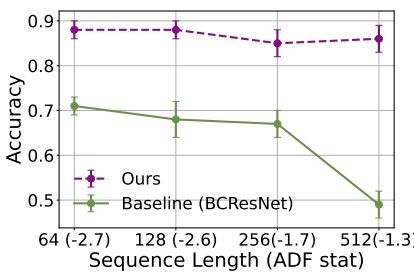

Figure 3: Comparison between PhASER (Ours) and BCResNet with increasingly nonstationary HHAR dataset.

The central question is: *What is the impact of the non-stationarity of time series on models' generalization ability?* We create a simple empirical study on the HHAR dataset and update the sequence length to build various levels of non-stationarity, which is measured by the Augmented Dickey-Fuller (ADF) statistics (a higher ADF value indicates greater non-stationarity) (Said & Dickey, 1984). More details of the ADF test are provided in Section B of the Appendix. We adopt Kim et al. (2021a)'s DG model, BCResNet, for time-series classification to explore the relationship between the degree of non-stationarity and the model's generalization ability to unseen domains. Figure 3 shows an evident drop in the accuracy of BCResNet as the non-stationarity increases, highlighting the *importance of addressing non-stationarity for achieving better generalization*. In contrast, our proposed PhASER framework, as detailed below, consistently performs well despite increasing non-stationarity.

**Overview of PhASER.** As shown before in Figure 1, our proposed PhASER framework begins with an augmentation module that utilizes the Hilbert Transform to generate out-of-phase augmentations for time series. These augmentations not only diversify non-stationarity (temporal data statistics) but also preserve category-discriminatory semantics for classification tasks. Next, the short-term Fourier Transform (STFT) is employed to obtain temporal magnitude and phase responses. Two

separate encoders then process the magnitude and phase as distinct input modalities. Finally, `PhASER` establishes a novel feature broadcasting mechanism to incorporate the phase information deeper in the layers through residual connections. By fully leveraging the phase-related information, the `PhASER` framework implicitly regularizes the representations against non-stationarity and offsets any degradation to the desirable features. Consequently, the classifier learns domain-agnostic task-discriminatory representations. In the following Sections 2.2 to 2.4, we will introduce the details of these three novel elements in `PhASER`, and then discuss the theoretical insights that inspire our design in Section 2.5.

## 2.2 HILBERT TRANSFORM BASED PHASE AUGMENTATION

Our motivating study depicted in Figure 3 demonstrated the importance of addressing non-stationarity to enhance the generalization ability of models. An intuitive direction is to leverage data augmentation to diversify the non-stationarity of training data. The optimal augmentation also needs to preserve the discriminatory properties of the original data, which is essential for semantic differentiability.

Unlike most existing time-series augmentation techniques, we introduce a phase shift to a signal while preserving the magnitude response, thereby offering an augmented view. This intra-sample phase-augmentation technique is less studied in the context of time-series classification for domain generalization (although some recent works like Demirel & Holz (2024) explore phase-mixup for contrastive learning), we intuitively justify our design choice by exploring a question: *Does shifting the phase of time-series spectral response change its non-stationarity?* Figure 1. I shows the result of accurately shifting the phase of a nonstationary signal without altering the magnitude response in the time domain and we can observe evident diversification of the non-stationarity statistics.

We propose a simple but effective data augmentation technique based on the Hilbert Transform (HT) to diversify the non-stationarity and preserve discriminatory features. Specifically, for each time-series sample $\mathbf{x}$ in the source dataset $\mathbf{S}$, we can assume it is a real-valued signal $\mathbf{x} = \{x_0, ..., x_t, ...\} \in \mathbb{R}$ that is characterized by a deterministic function $x_t = \mathbf{x}(t)$. Then, $\mathrm{HT}(\mathbf{x}(t)) = \widehat{\mathbf{x}}(t) = \int_{-\infty}^{\infty} \mathbf{x}(\tau) \frac{1}{\pi(t-\tau)} d\tau$. HT can be easily interpreted in the frequency domain via Fourier analysis:

$$f_{\mathbf{x}}(\xi) = \mathcal{F}\{\mathbf{x}(t)\} = \int_{-\infty}^{\infty} \mathbf{x}(t) e^{i2\pi\xi t} dt, -\infty < \xi < \infty,$$

$$\mathbf{x}(t) = \mathcal{F}^{-1}\{f_{\mathbf{x}}(\xi)\} = \int_{-\infty}^{\infty} f_{\mathbf{x}}(\xi) e^{i2\pi\xi t} d\xi, -\infty < t < \infty,$$

where $\mathcal{F}, \mathcal{F}^{-1}$ denote the Fourier transform and inverse, and $\xi$ is the frequency variable. To interpret $\widehat{\mathbf{x}}$ in the frequency domain, the negative frequency spectrum of $f_{\mathbf{x}}(\xi)$ needs to multiply with the imaginary unit $i$, while the positive spectrum needs to multiple with $-i$. Then we have:

$$\mathrm{HT}(\mathbf{x}(t)) = \widehat{\mathbf{x}}(t) = \mathcal{F}^{-1}\{-i \cdot \mathrm{sgn}(\xi) f_{\mathbf{x}}(\xi)\}, \tag{4}$$

where $\mathrm{sgn}(\cdot)$ is a sign function. Applying HT on a signal results in a phase shift of $-\pi/2$, yielding a new out-of-phase signal. After obtaining the transformed $\widehat{\mathbf{x}}$ for across all feature dimensions, we merge the augmented dataset $\widehat{\mathbf{S}}$ and the original $\mathbf{S}$ to form a new larger dataset $\mathbf{S}' = \widehat{\mathbf{S}} \cup \mathbf{S}$. For the rest of the design, there is no distinction among the samples in $\mathbf{S}'$, whether they belong to $\widehat{\mathbf{S}}$ or $\mathbf{S}$.

## 2.3 MAGNITUDE-PHASE SEPARATE ENCODING

After augmenting the source domain with phase-shift using HT, next, we identify optimal ways to encode time series for generalization. While employing spectral transformation is a common approach, our perspective diverges from most existing methods which typically focus on separating time and frequency information. Rather, we unify the time and frequency context, and instead consider the *magnitude* and *phase* information as distinct modalities of the original signals.

As we address the non-stationarity of time series, we adopt STFT rather than DFT. DFT is usually applicable to signals that are stationary and periodic over time, and not suitable for analyzing time-varying signals. STFT is obtained by applying DFT sequentially with a specified window through the entire length of the time series. Specifically, for each training sample $\mathbf{x} \in \mathbf{S}'$ with a continuous

time function $\mathbf{x}(t)$, sampling it at a fixed rate generates a discrete time series denoted as $\mathbf{x}[n]$ with a sequence length $N$, we have:

$$f_{\mathbf{x}}[n, k] = \sum_{m=n-(W-1)}^{n} w[n - m]\mathbf{x}[m]e^{i\xi_k m}. \tag{5}$$

The STFT of $\mathbf{x}[n]$, $f_{\mathbf{x}}[n, k]$, is a function of both discrete time $n$ and frequency bin indices $k$ with lengths $\widetilde{N}$ and $\Xi$, respectively. $\xi_k$ is a digital frequency variable given by $\xi_k = \frac{2\pi k}{\Xi}$ and $w[\cdot]$ is a window function. Without losing generality, we adopt the Hanning window with window length $W$, i.e., $w[n] = 0.5(1 - cos\frac{2\pi n}{W-1})$ where $0 \le n \le W - 1$. Note that the length and shape of the window determine the time-frequency resolution. A larger $W$ provides better frequency resolution and a smaller $W$ gives a better temporal scale. We set $W$ to be randomly sampled powers of 2 for each time-series feature, i.e., $W_i = 2^{p_i} \le \Xi, p_i \sim \mathcal{U} \in \mathbb{Z}_0^+, i \in [1, V]$, where $\mathcal{U}$ denotes a uniform distribution for integers. After obtaining $f_{\mathbf{x}}[n, k]$, we can compute its magnitude and phase as:

$$\mathrm{Mag}(\mathbf{x}) = \sqrt{\mathrm{Re}(f_{\mathbf{x}}[n, k])^2 + \mathrm{Im}(f_{\mathbf{x}}[n, k])^2}, \mathrm{Pha}(\mathbf{x}) = \arctan 2\left(\mathrm{Im}(f_{\mathbf{x}}[n]), \mathrm{Re}(f_{\mathbf{x}}[n, k])\right), \tag{6}$$

where $\mathrm{Im}(\cdot)$ and $\mathrm{Re}(\cdot)$ indicate imaginary and real parts of a complex number, and $\arctan 2(\cdot)$ is the two-argument form of arctan. Then we take $\mathrm{Mag}(\mathbf{x}), \mathrm{Pha}(\mathbf{x}) \in \mathbb{R}^{V \times \Xi \times \widetilde{N}}$ as inputs of two separate encoders $F_{\mathrm{Mag}}$ and $F_{\mathrm{Pha}}$. This approach is motivated by the viability of reconstructing a time-series signal using phase and magnitude responses (Hayes et al., 1980; Jacques & Feuillen, 2020), which is supported by our study below.

**Intuition of treating phase and magnitude as separate modalities.** Building on insights from prior studies (He et al., 2023; Kim et al., 2021a) highlighting the importance of spectral input in generalizable learning, we conduct a small-scale empirical study on the WISDM HAR dataset (Kwapisz et al., 2011) to explore optimal time-frequency input methods. Specifically, we compare four approaches: magnitude-only, phase-only, concatenated magnitude and phase, and separate encoders for magni-

Table 1: Comparison of various time-frequency input configurations.

| Input Modality | Accuracy |
|---|---|
| Only Magnitude (Mag) | $0.81 \pm 0.03$ |
| Only Phase (Pha) | $0.62 \pm 0.03$ |
| Mag-Pha Concatenate | $0.73 \pm 0.03$ |
| Mag-Pha Separate | $0.85 \pm 0.01$ |

tude and phase. Results (see Table 1) demonstrate that using only phase input yields inferior performance compared to magnitude-only input, suggesting the latter contains more discriminative information for classification tasks. Here the phase-only features achieve an accuracy of 0.62 in a six-class classification task – significantly higher than chance accuracy (0.17) – supporting the presence of task-discriminating but time-varying attributes in the phase response; motivating us to use it as an approximate proxy for signal's nonstationarity in `PhASER`. Also, concatenating magnitude and phase does not improve performance, whereas separate encoding followed by late fusion proves superior in this case. This may be attributed to 1) the independent selection of high-level features from the magnitude and phase for the task of classification, and 2) the learning about non-stationarity from the phase information.

Before fusing the extracted embeddings of $F_{\mathrm{Mag}}$ and $F_{\mathrm{Pha}}$, we incorporate sub-feature normalization proposed by Chang et al. (2021). Specifically, the embeddings of $F_{\mathrm{Mag}}$ and $F_{\mathrm{Pha}}$ are divided into $B$ sub-feature spaces. We apply normalization in each sub-feature space for each time-series variate, $F_{\mathrm{Mag}}(\mathbf{x}) = \left\{ F_{\mathrm{Mag}}(\mathbf{x})_b := \frac{F_{\mathrm{Mag}}(\mathbf{x})_b - \overline{F_{\mathrm{Mag}}(\mathbf{x})_b}}{\sigma(F_{\mathrm{Mag}}(\mathbf{x})_b)} \right\}_{b=1}^{B}$, where $\overline{(\cdot)}$ and $\sigma(\cdot)$ denote the computation of the mean and variance of the given input. The same sub-feature normalization is also conducted on $F_{\mathrm{Pha}}(\mathbf{x})$. Then, both $F_{\mathrm{Mag}}(\mathbf{x})$ and $F_{\mathrm{Pha}}(\mathbf{x})$ are fused along the variate axis by multiplying with 2D convolution kernels denoted as a fusing encoder $F_{\mathrm{Fus}}$. The fused embeddings $\mathbf{r}_{\mathrm{Fus}} = F_{\mathrm{Fus}}(F_{\mathrm{Mag}}(\mathbf{x}), F_{\mathrm{Pha}}(\mathbf{x}))$ are then fed into the following modules.

## 2.4 PHASE-RESIDUAL FEATURE BROADCASTING

Lastly, we outline our phase-based broadcasting approach to achieve domain generalizable representation learning. It starts with a depthwise feature encoder, $F_{\mathrm{Dep}}$, which transforms the fused embeddings, $\mathbf{r}_{\mathrm{Fus}}$, into 1D feature maps, $\mathbf{r}_{\mathrm{Dep}}$, along the temporal dimension, given as:

$$\mathbb{R}^{C(\mathbf{r}_{\mathrm{Fus}}) \times D(\mathbf{r}_{\mathrm{Fus}}) \times T(\mathbf{r}_{\mathrm{Fus}})} \to \mathbb{R}^{C(\mathbf{r}_{\mathrm{Fus}}) \times 1 \times T(\mathbf{r}_{\mathrm{Fus}})},$$

where $C(\cdot)$, $D(\cdot)$, and $T(\cdot)$ represent the channel number, the feature dimensions, and the temporal dimensions of an embedding. $F_{\text{Dep}}$ is implemented as several convolution layers followed by an average pooling operation to unify all features at each temporal index. Once the 1D feature map is obtained, we attach a sequence-to-sequence (the dimension format of the feature map remains intact) temporal encoder, $F_{\text{Tem}}$, to characterize its temporal dependency and semantics. The choice of backbone for $F_{\text{Tem}}$ is not central to our design and a suitable sequence-to-sequence encoder can be chosen. Here we leverage convolution layers to form $F_{\text{Tem}}$, and we have also tested other architectures (please refer to Section B in the Appendix for details). We adopt this feature consolidation approach to enable specialized learning of spectral attributes by $F_{\text{Dep}}$ and global temporal dependencies using $F_{\text{Tem}}$, resulting in a more valuable overall semantic characterization.

We now introduce a non-linear projection of $F_{\text{Pha}}(\mathbf{x})$ as a shortcut through $F_{\text{Dep}}$ to $F_{\text{Tem}}$. To suitably broadcast with the output dimensions of $F_{\text{Tem}}$, we use a projection head, $g_{\text{Res}}$ for the transformation:

$$\mathbb{R}^{C(F_{\text{Pha}}(\mathbf{x})) \times D(F_{\text{Pha}}(\mathbf{x})) \times T(F_{\text{Pha}}(\mathbf{x}))} \rightarrow \mathbb{R}^{C(\mathbf{r}_{\text{Fus}}) \times D(F_{\text{Pha}}(\mathbf{x})) \times T(\mathbf{r}_{\text{Fus}})}.$$

After the projection, we can broadcast the output of $F_{\text{Tem}}$ to form the final representation $\mathbf{r}$ that is intended to learn discriminatory characteristics despite non-stationarity:

$$\mathbf{r} = F_{\text{Tem}}(\mathbf{r}_{\text{Dep}}) + g_{\text{Res}}(F_{\text{Pha}}(\text{Pha}(\mathbf{x}))). \tag{7}$$

After these efforts to preserve and enhance the discriminatory characteristics amid input's non-stationarity, we now optimize for semantic distinction. This optimization is achieved with a Cross-Entropy Loss applied to a classification head $g_{\text{Cls}}$, which is attached to $F_{\text{Tem}}$ as $\mathcal{L}_{\text{CE}} = \frac{1}{N_B} \sum_{i=1}^{N_B} \mathbf{y}_i \log g_{\text{Cls}}(\mathbf{r})$, where $N_B$ is the size of a batch in the mini-batch training, and $\mathbf{y}_i$ is the one-hot form of the label $y_i$.

## 2.5 THEORETICAL INSIGHTS

Here we provide some theoretical insights to demonstrate that our method design is rigorously motivated. Detailed definitions and proofs are provided in Section A of the Appendix.

**Definition 2.3 ($\beta$-Divergence).** Suppose two data domains $\mathcal{D}_1$, $\mathcal{D}_2$ are built on input variable $\mathbf{x}$ and label variable $y$. Let $q > 0$ be a constant. The $\beta$-Divergence between $\mathcal{D}_1$ and $\mathcal{D}_2$ is defined as:

$$\beta_q(\mathcal{D}_1 \| \mathcal{D}_2) = \left[ \mathbb{E}_{(\mathbf{x},y) \sim \mathcal{D}_2} \left( \frac{\mathcal{D}_1(\mathbf{x}, y)}{\mathcal{D}_2(\mathbf{x}, y)} \right)^q \right]^{\frac{1}{q}}. \tag{8}$$

Per the definition in (Germain et al., 2016), $\beta$-Divergence can be linked to the Rényi Divergence (Van Erven & Harremos, 2014) $\text{RD}_q(\cdot)$ as:

$$\beta_q(\mathcal{D}_1 \| \mathcal{D}_2) = 2^{\frac{q-1}{q} \text{RD}_q(\mathcal{D}_1 \| \mathcal{D}_2)}. \tag{9}$$

**Lemma 2.4 (Bounding $\beta$-Divergence in A Convex Hull).** *Let $S$ be a set of source domains, denoted as $S = \{\mathcal{S}_i\}_{i=1}^{N_S}$. A convex hull $\Lambda_S$ considered here consists of a mixture distributions $\Lambda_S = \{\bar{\mathcal{S}} : \bar{\mathcal{S}}(\cdot) = \sum_{i=1}^{N_S} \pi_i \mathcal{S}_i(\cdot), \pi_i \in \Delta_{N_S-1}\}$, where $\Delta_{N_S-1}$ is the $(N_S-1)$-th dimensional simplex. Let $\beta_q(\mathcal{S}_i \| \mathcal{S}_j) \leq \epsilon$ for $\forall i, j \in [N_S]$, and then we have the following relation for the $\beta$-Divergence between any pair of two domains $\mathcal{D}', \mathcal{D}'' \in \Lambda_S$ in the convex hull:*

$$\beta_q(\mathcal{D}' \| \mathcal{D}'') \leq \epsilon. \tag{10}$$

**Theorem 2.5 (Risk of An Unseen Time-Series Domain).** *Let $\mathcal{H}$ be a hypothesis space built from a set of source time-series domains, denoted as $S = \{\mathcal{S}_i\}_{i=1}^{N_S}$ with the same value range (i.e., the supports of these source domains are the same). Suppose $q > 0$ is a constant. For any unseen time-series domain $\mathcal{D}_{\text{U}}$ from the convex hull $\Lambda_S$, we have its closest element $\mathcal{D}_{\bar{\text{U}}}$ in $\Lambda_S$, i.e., $\mathcal{D}_{\bar{\text{U}}} = \arg \min_{\pi_1, \dots, \pi_{N_S}} \beta_q(\mathcal{D}_{\bar{\text{U}}} \| \sum_{i=1}^{N_S} \pi_i \mathcal{S}_i)$. Then the risk of $\mathcal{D}_{\text{U}}$ on any $\rho$ in $\mathcal{H}$ is:*

$$R_{\mathcal{D}_{\text{U}}}[\rho] \leq \frac{1}{2} \text{d}_{\mathcal{D}_{\text{U}}}(\rho) + \epsilon \cdot \left[ \text{e}_{\mathcal{D}_{\bar{\text{U}}}}(\rho) \right]^{1 - \frac{1}{q}}, \tag{11}$$

*where $\text{d}_{\mathcal{D}}(\rho)$ and $\text{e}_{\mathcal{D}}(\rho)$ are an expected disagreement and an expected joint error of a domain $\mathcal{D}$, respectively. The $\epsilon$ is a value larger than the maximum $\beta$-Divergence in $\Lambda_S$:*

$$\epsilon \geq \max_{i,j \in [N_S], i \neq j, t \in [0, +\infty)} 2^{\frac{q-1}{q} \text{RD}_q(\mathcal{S}_i(t) \| \mathcal{S}_j(t))}, \tag{12}$$

$$where \ \text{RD}_q(\mathcal{S}_i(t) \| \mathcal{S}_j(t)) = \frac{q(\mu_{j,t} - \mu_{i,t})^2}{2(1-q)\sigma_{i,t}^2 + 2\sigma_{j,t}^2} + \frac{\ln \frac{\sqrt{(1-q)\sigma_{i,t}^2 + \sigma_{j,t}^2}}{\sigma_{i,t}^{1-q}\sigma_{j,t}^q}}{1-q}. \tag{13}$$

**Insights.** Theorem 2.5 indicates potential efforts to reduce the generalization risk of an unseen target domain. According to Eq. (11), the risk is bounded by two terms. The first term $d_{\mathcal{D}_U}(\rho)$ is the expected disagreement of $\mathcal{D}_U$ and we are unable to conduct any approximation without accessing the data from $\mathcal{D}_U$. Regarding the second term, the coefficient $\epsilon$ can be viewed as the maximum $\beta$-Divergence of source domains, and according to Eq. (13), the nonstationary statistics of time series are arguments of the $\beta$-Divergence. We regard the $\beta$-Divergence as a proxy for non-stationarity. However, since directly approximating it in the raw feature space is infeasible, we instead approximate the $\beta$-Divergence in the representation space. Specifically, we perform this approximation at two levels: the low-level representation space extracted by the phase feature encoder $F_{\text{Pha}}$ and the high-level representation space extracted by the temporal feature encoder $F_{\text{Tem}}$. To effectively minimize these approximations, we introduce a residual connection that links these two levels of representation, facilitating a better alignment and reduction of non-stationarity. Besides, $e_{\mathcal{D}_{\tilde{U}}}(\rho)$ shows that the empirical risks of source domains need to be minimized. Such insights are well reflected in PhASER.

**Theorem 2.6 (Non-stationarity Change of Hilbert Transform).** *Suppose there are $M_{\mathcal{D}}$ samples (observations) available for a nonstationary time-series domain $\mathcal{D}_{\mathbf{x}}$, and each sample $\mathbf{x}_i = \{x_{i,0}, ..., x_{i,t}, ...\}$ is characterized by its deterministic function, i.e., $\mathbf{x}_i(t) = x_{i,t} = \mathrm{x}_i(t)$, $i \in [1, M_{\mathcal{D}}]$. If we apply Hilbert Transformation $\mathrm{HT}(\mathbf{x}(t)) = \hat{\mathbf{x}}(t) = \int_{-\infty}^{\infty} \mathrm{x}(\tau) \frac{1}{\pi(t-\tau)} d\tau$ to augment these time-series samples, the nonstationary statistics of augmented samples are different from the original ones, $\mathrm{Pr}_{\mathbf{x} \sim \widehat{\mathcal{D}}_{\mathbf{x}}}(\mathbf{x})(t) \neq \mathrm{Pr}_{\mathbf{x} \sim \mathcal{D}_{\mathbf{x}}}(\mathbf{x})(t)$.*

**Insights.** This theorem illustrates that HT does change the nonstationary statistics of time series, proving that our phase augmentation can diversify the non-stationary of time series.

## 3 EXPERIMENTS

We extensively evaluate our proposed PhASER framework against 13 state-of-the-art approaches (including a large foundation time-series model), on 5 datasets across three time-series applications. Our evaluation metric is per-segment accuracy. More implementation-specific details are provided in Section D of the Appendix. Our source codes are provided in the Supplementary Materials.

**Datasets.** We conduct experiments on three common time-series applications – Human Activity Recognition (HAR), Sleep-Stage Classification (SSC), and Gesture Recognition (GR). For HAR, we use 3 benchmark datasets: **WISDM** (Kwapisz et al., 2011) collected from 36 different users with 3 univariate dimensions, **UCIHAR** (Bulbul et al., 2018) collected from 30 people with 9 variates, and **HHAR** (Stisen et al., 2015) collected from 9 users with 3 feature dimensions, comprising 6 distinct activities with a sequence length of 128. For SSC, the dataset (Goldberger et al., 2000) consists of single-channel EEG data from 20 healthy individuals with a sequence length of 3000. For GR, the dataset (Lobov et al., 2018) is 8-channel EMG data for 6 different gestures, with a sequence length of 200, prepared similarly as in (Lu et al., 2022b). We follow the setup of ADATime (Ragab et al., 2023a) for HAR and SSC. More data-specific details are provided in Table 8 of the Appendix.

**Experimental Setup.** Each dataset is divided into four distinct non-overlapping cross-domain scenarios, following the approach in (Lu et al., 2023). Details are provided in Section D.1 of the

Table 2: Classification accuracy of Target 1∼4 scenarios for cross-person generalization in Human Activity Recognition on WISDM, HHAR, and UCIHAR (**Best** in bold, second-best underlined).

| Dataset | WISDM | | | | | HHAR | | | | | UCIHAR | | | | | HAR |
|---|---|---|---|---|---|---|---|---|---|---|---|---|---|---|---|---|
| Target | 1 | 2 | 3 | 4 | Avg. | 1 | 2 | 3 | 4 | Avg. | 1 | 2 | 3 | 4 | Avg. | Avg. |
| ERM | 0.57 | 0.50 | 0.51 | 0.55 | 0.53 | 0.49 | 0.46 | 0.45 | 0.47 | 0.47 | 0.72 | 0.64 | 0.70 | 0.72 | 0.70 | 0.57 |
| GroupDRO | 0.71 | 0.67 | 0.60 | 0.67 | 0.66 | 0.60 | 0.53 | 0.59 | 0.64 | 0.59 | 0.91 | 0.84 | 0.89 | 0.85 | 0.87 | 0.71 |
| DANN | 0.71 | 0.65 | 0.65 | 0.70 | 0.68 | 0.66 | 0.71 | 0.67 | 0.69 | 0.68 | 0.84 | 0.79 | 0.81 | 0.86 | 0.83 | 0.73 |
| RSC | 0.69 | 0.71 | 0.64 | 0.61 | 0.66 | 0.52 | 0.49 | 0.44 | 0.47 | 0.48 | 0.82 | 0.73 | 0.74 | 0.81 | 0.78 | 0.64 |
| ANDMask | 0.74 | 0.73 | 0.69 | 0.69 | 0.71 | 0.63 | 0.64 | 0.66 | 0.69 | 0.66 | 0.86 | 0.80 | 0.76 | 0.78 | 0.80 | 0.72 |
| InceptionTime | 0.83 | 0.82 | 0.80 | 0.77 | 0.81 | 0.77 | 0.80 | 0.82 | 0.83 | 0.80 | 0.91 | 0.82 | 0.88 | 0.91 | 0.88 | 0.82 |
| BCResNet | 0.83 | 0.79 | 0.75 | 0.78 | 0.79 | 0.66 | 0.70 | 0.75 | 0.68 | 0.70 | 0.81 | 0.77 | 0.78 | 0.83 | 0.80 | 0.76 |
| NSTrans | 0.43 | 0.40 | 0.37 | 0.37 | 0.40 | 0.21 | 0.22 | 0.27 | 0.28 | 0.24 | 0.35 | 0.35 | 0.51 | 0.47 | 0.42 | 0.35 |
| Koopa | 0.63 | 0.61 | 0.72 | 0.57 | 0.63 | 0.72 | 0.63 | 0.72 | 0.69 | 0.69 | 0.81 | 0.72 | 0.81 | 0.77 | 0.78 | 0.70 |
| MAPU | 0.75 | 0.69 | 0.79 | 0.79 | 0.75 | 0.73 | 0.72 | 0.81 | 0.78 | 0.76 | 0.85 | 0.80 | 0.85 | 0.82 | 0.83 | 0.78 |
| Diversify | 0.82 | 0.82 | 0.84 | 0.81 | 0.82 | 0.82 | 0.76 | 0.82 | 0.68 | 0.77 | 0.89 | 0.84 | 0.93 | 0.90 | 0.89 | 0.83 |
| Chronos | 0.71 | 0.66 | 0.65 | 0.62 | 0.66 | 0.66 | 0.73 | 0.75 | 0.66 | 0.72 | 0.56 | 0.57 | 0.50 | 0.82 | 0.61 | 0.67 |
| Ours+RevIN* | 0.86 | 0.85 | 0.84 | 0.84 | 0.85 | 0.82 | 0.82 | 0.92 | 0.85 | 0.85 | 0.96 | 0.90 | 0.93 | 0.97 | 0.94 | 0.88 |
| Ours | 0.86 | 0.85 | 0.85 | 0.82 | 0.85 | 0.83 | 0.83 | 0.94 | 0.88 | 0.87 | 0.96 | 0.91 | 0.95 | 0.97 | 0.95 | 0.89 |

Table 3: Classification accuracy with Source 0∼8 person for one-person-to-another generalization on the HHAR dataset (**Best** in bold, second-best underlined).

| Source | 0 | 1 | 2 | 3 | 4 | 5 | 6 | 7 | 8 | Avg. |
|---|---|---|---|---|---|---|---|---|---|---|
| ERM | 0.27 | 0.40 | 0.41 | 0.44 | 0.42 | 0.44 | 0.45 | 0.44 | 0.48 | 0.42 |
| GroupDRO | 0.33 | 0.53 | 0.38 | 0.48 | 0.47 | 0.51 | 0.47 | 0.48 | 0.49 | 0.46 |
| DANN | 0.32 | 0.44 | 0.42 | 0.45 | 0.42 | 0.48 | 0.49 | 0.45 | 0.51 | 0.44 |
| RSC | 0.27 | 0.45 | 0.38 | 0.45 | 0.40 | 0.47 | 0.50 | 0.44 | 0.53 | 0.43 |
| ANDMask | 0.34 | 0.50 | 0.37 | 0.43 | 0.46 | 0.51 | 0.46 | 0.47 | 0.52 | 0.45 |
| InceptionTime | 0.52 | 0.62 | 0.44 | **0.69** | 0.60 | 0.57 | 0.66 | 0.64 | 0.61 | 0.59 |
| BCResNet | 0.28 | 0.48 | 0.32 | 0.47 | 0.42 | 0.52 | 0.44 | 0.45 | 0.49 | 0.43 |
| NSTrans | 0.20 | 0.22 | 0.17 | 0.20 | 0.21 | 0.22 | 0.26 | 0.17 | 0.20 | 0.21 |
| Koopa | 0.32 | 0.42 | 0.37 | 0.40 | 0.42 | 0.45 | 0.35 | 0.43 | 0.48 | 0.40 |
| MAPU | 0.39 | 0.57 | 0.35 | 0.52 | 0.49 | 0.54 | 0.49 | 0.50 | 0.52 | 0.49 |
| Diversify | 0.42 | 0.62 | 0.32 | 0.62 | 0.56 | 0.61 | 0.53 | 0.52 | 0.61 | 0.53 |
| Chronos | 0.32 | 0.23 | 0.26 | 0.25 | 0.27 | 0.23 | 0.21 | 0.24 | 0.25 | 0.25 |
| Ours | **0.53** | **0.70** | **0.63** | 0.66 | **0.64** | **0.67** | **0.65** | **0.67** | **0.62** | **0.64** |

Table 4: Classification accuracy for cross-person generalization (Target 1∼4) Sleep-Stage Classification (EEG) and Gesture Recognition (EMG) (**Best** in bold, second-best underlined).

| Application | Sleep-Stage Classification | | | | | Gesture Recognition | | | | |
|---|---|---|---|---|---|---|---|---|---|---|
| Target | 1 | 2 | 3 | 4 | Avg. | 1 | 2 | 3 | 4 | Avg. |
| ERM | 0.50 | 0.46 | 0.49 | 0.45 | 0.47 | 0.45 | 0.58 | 0.57 | 0.54 | 0.54 |
| GroupDRO | 0.57 | 0.56 | 0.55 | 0.59 | 0.57 | 0.53 | 0.36 | 0.59 | 0.45 | 0.48 |
| DANN | 0.64 | 0.63 | 0.69 | 0.63 | 0.65 | 0.60 | 0.66 | 0.65 | 0.64 | 0.64 |
| RSC | 0.50 | 0.48 | 0.52 | 0.46 | 0.49 | 0.50 | 0.66 | 0.64 | 0.56 | 0.59 |
| ANDMask | 0.55 | 0.50 | 0.54 | 0.57 | 0.54 | 0.41 | 0.54 | 0.45 | 0.39 | 0.45 |
| InceptionTime | 0.74 | 0.78 | 0.72 | 0.80 | 0.76 | 0.68 | 0.70 | 0.72 | 0.69 | 0.70 |
| BCResNet | 0.79 | 0.82 | 0.79 | 0.81 | 0.80 | 0.62 | 0.67 | 0.65 | 0.61 | 0.64 |
| NSTrans | 0.43 | 0.37 | 0.42 | 0.35 | 0.39 | 0.31 | 0.34 | 0.34 | 0.32 | 0.33 |
| Koopa | 0.58 | 0.62 | 0.53 | 0.49 | 0.56 | 0.47 | 0.54 | 0.60 | 0.70 | 0.58 |
| MAPU | 0.69 | 0.68 | 0.65 | 0.69 | 0.68 | 0.64 | 0.69 | 0.71 | 0.68 | 0.68 |
| Diversify | 0.73 | 0.76 | 0.68 | 0.77 | 0.73 | 0.68 | 0.80 | 0.75 | **0.76** | 0.75 |
| Chronos | 0.53 | 0.47 | 0.47 | 0.57 | 0.51 | 0.49 | 0.54 | 0.51 | 0.48 | 0.51 |
| Ours | **0.85** | **0.80** | **0.79** | **0.83** | **0.82** | **0.70** | **0.82** | **0.77** | 0.75 | **0.76** |

Appendix. 20% of the training data is reserved for validation. Mean results from three trials are reported in the main text, with full statistics in Section E of the Appendix.

**Comparison Baselines.** We conduct comparison with state-of-the-art approaches including domain generalization algorithms – ERM, DANN (Ganin et al., 2016), GroupDRO (Sagawa et al., 2019), RSC (Huang et al., 2020) and ANDMask (Parascandolo et al., 2020) implemented based on the DomainBed benchmarking suite (Gulrajani & Lopez-Paz, 2020); an audio domain generalization method BCResNet (Kim et al., 2021b); a time-series representation learning method MAPU (Ragab et al., 2023b); a strong deep-learning time-series classification model (top ranked by Middlehurst et al. (2024)), InceptionTime (Ismail Fawaz et al., 2020), a time-series domain generalizable learning method Diversify (Lu et al., 2022b); and a large time-series foundation model Chronos (Ansari et al., 2024). We also adapt the time-series forecasting models Nonstationary Transformer (NSTrans) (Liu et al., 2022) and Koopa (Liu et al., 2024), and integrate a network-agnostic statistical technique RevIN (Kim et al., 2021c) with our method (denoted as Ours+RevIN*). We follow the default setups of these works and only conduct necessary modification for our problem setting. Details are in Sections D.2 and D.6 of the Appendix.

## 3.1 EFFECTIVENESS OF PHASER ACROSS APPLICATIONS

**Human Activity Recognition.** We assess the generalization ability of PhASER framework in two settings: 1) *cross-person generalization*, where the model is trained on $N_S$ ($N_S > 1$) source domains and evaluated on unseen target domains, and 2) *one-person-to-another*, where the model is trained on one person ($N_S = 1$) and evaluated on another person. In the cross-person setting, as shown in Table 2, we find that existing state-of-the-art domain generalization methods, popular in vision-based domains, do not perform as well in time-series classification (such observation is consistent with previous works (Gagnon-Audet et al., 2022; Lu et al., 2022b)). **PhASER achieves superior out-of-domain generalization performance across all cases, notably outperforming the best baseline on WISDM, HHAR, and UCIHAR by 3%, 9%, and 6%, respectively**. In the more challenging one-person-to-another setting, as shown in Table 3, we select the HHAR dataset due to its high non-stationarity, and the results show that **PhASER excels in this setting as well, outperforming Diversify by almost 20% and InceptionTime by almost 8%**.

**Sleep-Stage Classification.** Next, we evaluate PhASER for *cross-person generalization* in five types of sleep-stage classification using EEG. Past methods (Ragab et al., 2023a; He et al., 2023) generally report the lowest performance in their respective settings for SSC tasks indicating its inherent complexity. The results in Table 4 (left) show that **PhASER provides the best performance in all cases, outperforming the best baseline (BCResNet) by 2% and the time-series domain generalization baseline (Diversify) by almost 11%**.

**Gesture Recognition.** In GR, the used bio-electronic signals are heavily influenced by user behavior and sensor time-varying properties, which correspond to natural non-stationarity. We follow the approach in (Lu et al., 2023) to use 6 common classes when conducting evaluations in a *cross-person setting*. The results in Table 4 (right) show that **PhASER again offers the best overall performance**.

| | Phase Augmentation | Separate Encoders | $F_{\text{Pha}}$ Residual | Accuracy WISDM | Accuracy GR |
|---|---|---|---|---|---|
| 1 | ✓ | ✓ | ✓ | $0.86_{\pm0.02}$ | $0.70_{\pm0.01}$ |
| 2 | ✗ | ✓ | ✓ | $0.81_{\pm0.01}$ | $0.61_{\pm0.01}$ |
| 3 | ✓ | ✓ | ✗($F_{\text{Mag}}$ Res.) | $0.82_{\pm0.01}$ | $0.55_{\pm0.01}$ |
| 4 | ✓ | ✓ | ✗($F_{\text{Fus}}$ Res.) | $0.84_{\pm0.01}$ | $0.60_{\pm0.01}$ |
| 5 | ✓ | ✓ | ✗ | $0.82_{\pm0.01}$ | $0.65_{\pm0.01}$ |
| 6 | ✓ | ✗(Mag Only) | ✗ | $0.73_{\pm0.01}$ | $0.59_{\pm0.03}$ |
| 7 | ✓ | ✗(Mag Only) | ✗($F_{\text{Mag}}$ Res.) | $0.83_{\pm0.01}$ | $0.66_{\pm0.02}$ |

Table 5: Ablation of PhASER on WISDM and GR. The inclusion of a component is denoted as ✓ and exclusion as ✗ (modification).

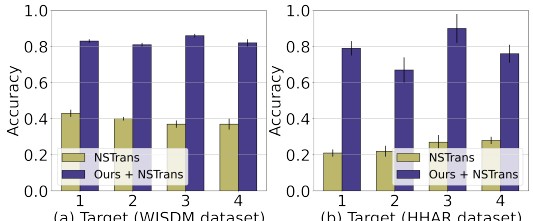

Figure 4: Improvement in average cross-person generalization performance of NSTrans in (a) WISDM from 0.40 to 0.83 and (b) HHAR from 0.25 to 0.78, with our phase-driven approach.

## 3.2 FURTHER ANALYSIS

**Ablation Study.** We examine the impact of our proposed design components in two cases: WISDM and GR (Table 5). The first row represents the performance of the complete PhASER framework, with subsequent rows showing performance with specific components detached or modified (details in Section D of the Appendix). When phase augmentation is omitted (row 2), performance notably decreases (by 11.6% on WISDM and 5.8% on GR). Comparing the results of row 6 with that of row 5 confirms the importance of separate phase-magnitude encoding, aligning with findings from our motivation study in Table 1. Under identical conditions (comparing row 5 with row 1), phase-residual broadcasting boosts the performance of PhASER by 4%, aligning well with our design motivation that phase can be considered a proxy for non-stationarity. Reintroducing this phase-dictionary deeper in the layers enables the model to learn task-specific representations that are more robust to non-stationarity, making it better equipped to handle unseen non-stationarity in the target domains. Removing the phase-based residual and separate encoding structure (rows 3-7 in Table 5) results in average performance drops of 10.6% and 13.7%, respectively. **This demonstrates the value of all the components in PhASER**.

**General Applicability of PhASER.** Table 2 shows that existing time-series classification models like RevIN can be seamlessly integrated into PhASER and achieve good results (also see Tables 13 and 14 in the Appendix). Moreover, we demonstrate the general applicability and flexibility of PhASER by incorporating three proposed design elements into the NSTrans model for classification: phase-based augmentation for non-stationarity diversification, separate magnitude-phase feature encoding, and phase incorporation with a residual connection. Significant performance improvements on WISDM and HHAR (Figure 4) highlight the effectiveness of these designs and the flexibility of PhASER with different backbone models. Further details are provided in Section D.4 of the Appendix.

**PhASER with Other Augmentation Strategies.** Here, we explore a random phase augmentation-variant using Hilbert Transform under certain signal periodicity assumptions (more details in Section D.7.2 in the Appendix). Additionally, we adopt traditional augmentations like rotation, permutation, and circular time-shift as proposed by past works (Qin et al., 2023; Um et al., 2017) on the HHAR dataset with the PhASER framework. The results are illustrated in Figure 5 and implementation details are provided in Section D.7 of the Appendix. The rotation and permutation augmentations perform 5% worse than the no augmentation scenario in this case. Time-shift may be viewed as a linear phase shift for a pure sinusoid (for example, for an input

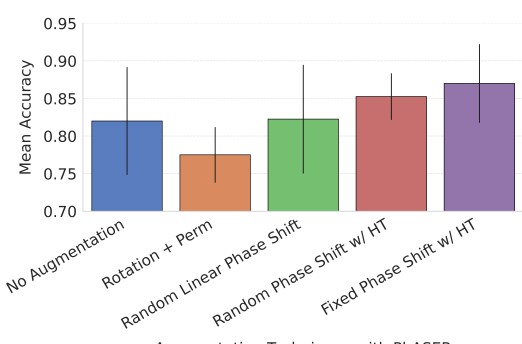

Figure 5: A brief comparison between different augmentation strategies with PhASER.

$\mathbf{x}(t) = sin(\omega t)$, a time-shifted version by $T$ time units is given by $\mathbf{x}(t - T) = sin(\omega(t - T))$ which incurs a phase shift $\phi = \omega T$), however, most real-world signals are not stationary or pure tone. In such a case, a time shift introduces varied phase shifts for each frequency, and past works like Umapathy et al. (2010) expose the difficulty in the correct choice of a time-shift amount for retaining the signal's spectral properties of interest. This highlights the superiority of Hilbert Transform to provide an accurate phase shift of all frequency components by $-\pi/2$ without any explicit signal

characterization. Our exploration to induce random phase shift using HT does not show any particular advantage, hence we stick to the choice of using the fixed phase-shift augmentation followed by other phase-anchored components for domain generalization in nonstationary time-series classification tasks in the proposed `PhASER` framework.

**Visualization.** We provide t-SNE visualizations of our method (`PhASER`), Diversify, and BCResNet on the HHAR dataset for left-out domains in scenario 1 (Figure 6). The plots depict out-of-domain data, with colors representing the six activity classes, showcasing `PhASER`'s superior separability without domain labels or target domain data. Further details are in Section D.8 of the Appendix.

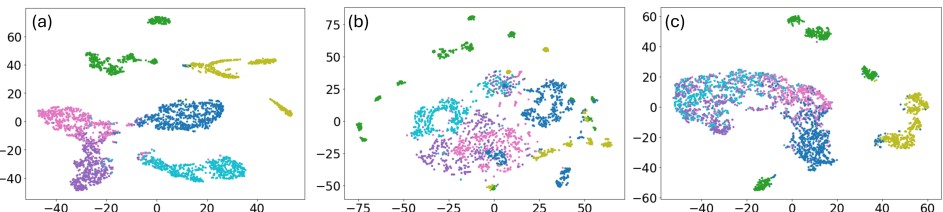

Figure 6: t-sne visualization for (a) `PhASER`, (b) Diversify, and (c) BCResNet for HHAR scenario 1.

## 4 RELATED WORKS

**Nonstationary Time-Series Analysis.** In real-world scenarios, nonstationary time-series data pose challenges for forecasting and classification (Esling & Agon, 2012; Ismail Fawaz et al., 2019; Dama & Sinoquet, 2021). While various solutions exist, including Bayesian models, normalization techniques, recurrent neural networks, and transformers, systematic works addressing non-stationarity's impact on time-series classification are limited (Liang, 2005; Chen & Sun, 2021; Liu et al., 2023b; Chang et al., 2021; Passalis et al., 2019; Tang et al., 2021; Du et al., 2021; Liu et al., 2022; Wang et al., 2022a). Our study is the first to rigorously address the impact of non-stationarity on time-series out-of-distribution classification, complementing empirical findings from prior works (Zhao et al., 2020; Tonekaboni et al., 2020; Eldele et al., 2023).

**Domain Generalizable Learning.** While domain generalizable learning is well-established in visual data (Wang et al., 2022b), applying it to time-series data poses unique challenges. Traditional approaches like data augmentation (Wang et al., 2021) and domain discrepancy minimization (Zhang & Chen, 2023; Li et al., 2018) face limitations in time series due to less flexible augmentation and broader domain concepts (Wen et al., 2021; Wilson et al., 2020). Some studies explore domain-invariant representation learning (Lu et al., 2023; Wang et al., 2023) and learnable data transformation (Qin et al., 2023). We highlight the non-stationarity of time series and its role in domain discrepancy, drawing on evidence from the visual domain regarding the importance of phase (Kim et al., 2023; Xu et al., 2021). A handful of works hint at phase's role in domain-invariant learning in time-series applications (Lu et al., 2022a), and there is evidence in traditional signal processing that phase-only information is sufficient to reconstruct a signal (Masuyama et al., 2023; Jacques & Feuillen, 2020; 2021). Inspired by these insights, we propose a novel phase-driven framework with an augmentation module and a phase-anchored representation learning to address non-stationarity and minimize domain discrepancy.

## 5 LIMITATIONS AND FUTURE WORK

`PhASER` achieves domain generalization without explicit domain characterization or accessing target domain samples, by diversifying non-stationarity and anchoring design to signal phase information. Our evaluation is currently limited to categorical tasks due to a scarcity of publicly available datasets with distinct domain definitions for continuous tasks like regression. Our future work aims to develop a universal representation for generalization across various tasks in dynamic conditions.

## 6 CONCLUSION

We address the generalization problem for nonstationary time-series classification using a phase-driven approach without accessing domain labels of source domains or samples from unseen distributions. Our approach conducts phase-based augmentation, treats time-varying magnitude and phase as

separate modalities, and incorporates a phase-derived residual connection in the network. We support our design choices with rigorous theoretical and empirical evidence. Our method demonstrates significant improvement over baselines across 13 benchmarks on 5 real-world datasets.

## REPRODUCIBILITY STATEMENT

All source codes to reproduce experiment results (with instructions for running the code) have been provided in the Supplementary Materials. We use public datasets and provide implementation details in the following Appendix.

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

# APPENDIX

This Appendix includes additional details for the paper *"Phase-driven Domain Generalizable Learning for Nonstationary Time Series"*, including the reproducibility statement, theoretical proofs (Section A), additional details of `PhASER` (Section B), detailed dataset introduction (Section C), implementation details (Section D), and detailed results (Section E) of main experiments.

## A  THEORETICAL PROOFS

**Lemma** 2.4. *Let a set $S$ of source domains $S = \{\mathcal{S}_i\}_{i=1}^{N_S}$. A convex hull $\Lambda_S$ is considered here that consists of mixture distributions $\Lambda_S = \{\bar{\mathcal{S}} : \bar{\mathcal{S}}(\cdot) = \sum_{i=1}^{N_S} \pi_i \mathcal{S}_i(\cdot), \pi_i \in \Delta_{N_S-1}\}$, where $\Delta_{N_S-1}$ is the $(N_S-1)$-th dimensional simplex. Let $\beta_q(\mathcal{S}_i \| \mathcal{S}_j) \leq \epsilon$ for $\forall i, j \in [N_S]$, we have the following relation for the $\beta$-Divergence between any pair of two domains $\mathcal{D}'$, $\mathcal{D}'' \in \Lambda_S$ in the convex hull,*

$$\beta_q(\mathcal{D}' \| \mathcal{D}'') \leq \epsilon. \tag{14}$$

**Proof.** Suppose two unseen domains $\mathcal{D}'$ and $\mathcal{D}_{//}$ on the convex hull $\Lambda_S$ of $N_S$ source domains with support $\Omega$. More specifically, let these two domains be $\mathcal{D}' = \sum_{k=1}^{N_S} \pi_k \mathcal{S}_k(\cdot)$ and $\mathcal{D}'' = \sum_{l=1}^{N_S} \pi_l \mathcal{S}_l(\cdot)$, then the $\beta$-Divergence between $\mathcal{D}'$ and $\mathcal{D}''$ is

$$\beta_q(\mathcal{D}' \| \mathcal{D}'') = 2^{\frac{q-1}{q} \mathrm{RD}_q(\mathcal{D}' \| \mathcal{D}'')}. \tag{15}$$

Let us consider the part of Rényi Divergence as follows,

$$
\begin{aligned}
\mathrm{RD}_q(\mathcal{D}' \| \mathcal{D}'') &= \frac{1}{q-1} \ln \int_\Omega \left[\mathcal{D}'(x)\right]^q \left[\mathcal{D}''(x)\right]^{1-q} dx \\
&= \frac{1}{q-1} \ln \int_\Omega \left[\sum_{k=1}^{N_S} \pi_k \mathcal{S}_k(x)\right]^q \left[\sum_{l=1}^{N_S} \pi_l \mathcal{S}_l(x)\right]^{1-q} dx \\
&= \frac{1}{q-1} \ln \int_\Omega \left[\sum_{k=1}^{N_S}\sum_{l=1}^{N_S} \pi_k \pi_l \mathcal{S}_k(x)\right]^q \left[\sum_{k=1}^{N_S}\sum_{l=1}^{N_S} \pi_k \pi_l \mathcal{S}_l(x)\right]^{1-q} dx \\
&= \frac{1}{q-1} \ln \sum_{k=1}^{N_S}\sum_{l=1}^{N_S} \pi_k \pi_l \int_\Omega \left[\mathcal{S}_k(x)\right]^q \left[\mathcal{S}_l(x)\right]^{1-q} dx \\
&\leq \frac{1}{q-1} \ln \sum_{k=1}^{N_S}\sum_{l=1}^{N_S} \pi_k \pi_l \max_{k,l \in [N_S]} \int_\Omega \left[\mathcal{S}_k(x)\right]^q \left[\mathcal{S}_l(x)\right]^{1-q} dx \\
&= \frac{1}{q-1} \ln \max_{k,l \in [N_S]} \int_\Omega \left[\mathcal{S}_k(x)\right]^q \left[\mathcal{S}_l(x)\right]^{1-q} dx.
\end{aligned} \tag{16}
$$

According to the given assumption that $\beta_q(\mathcal{S}_i \| \mathcal{S}_j) \leq \epsilon$ for $\forall i, j \in [N_S]$, we have,

$$\mathrm{RD}_q(\mathcal{D}' \| \mathcal{D}'') \leq \frac{1}{q-1} \ln \max_{k,l \in [N_S]} \int_\Omega \left[\mathcal{S}_k(x)\right]^q \left[\mathcal{S}_l(x)\right]^{1-q} dx = \max_{k,l \in [N_S]} \mathrm{RD}_q(\mathcal{S}_k \| \mathcal{S}_l) \leq \frac{q}{q-1} \log_2 \epsilon. \tag{17}$$

Thus $\beta_q(\mathcal{D}' \| \mathcal{D}'') \leq \epsilon$. $\qquad\square$

**Theorem** 2.5. *Let $\mathcal{H}$ be a hypothesis space built from a set of source time-series domains $S = \{\mathcal{S}_i\}_{i=1}^{N_S}$ with the same value range (i.e., the supports of these source domains are the same). Suppose $q > 0$ is a constant, for any unseen time-series domain $\mathcal{D}_U$ from the convex hull $\Lambda_S$, we have its closest element $\mathcal{D}_{\bar{U}}$ in $\Lambda_S$, i.e., $\mathcal{D}_{\bar{U}} = \arg\min_{\pi_1,\dots,\pi_{N_S}} \beta_q(\mathcal{D}_{\bar{U}} \| \sum_{i=1}^{N_S} \pi_i \mathcal{S}_i)$. Then the risk of $\mathcal{D}_U$ on any $\rho$ in $\mathcal{H}$ is,*

$$R_{\mathcal{D}_U}[\rho] \leq \frac{1}{2} \mathrm{d}_{\mathcal{D}_U}(\rho) + \epsilon \cdot \left[\mathrm{e}_{\mathcal{D}_{\bar{U}}}(\rho)\right]^{1-\frac{1}{q}}, \tag{18}$$

*where $\mathrm{d}_{\mathcal{D}}(\rho)$ and $\mathrm{e}_{\mathcal{D}}(\rho)$ are an expected disagreement and an expected joint error of a domain $\mathcal{D}$, respectively, and they are defined as follows,*

$$\mathrm{d}_{\mathcal{D}}(\rho) = \mathbb{E}_{\mathbf{x}\sim\mathcal{D}_{\mathbf{x}}}\mathbb{E}_{h\sim\rho}\mathbb{E}_{h'\sim\rho}\mathrm{I}[h(\mathbf{x}) \neq h'(\mathbf{x})], \tag{19}$$

$$\mathrm{e}_{\mathcal{D}}(\rho) = \mathbb{E}_{(\mathbf{x},y)\sim\mathcal{D}}\mathbb{E}_{h\sim\rho}\mathbb{E}_{h'\sim\rho}\mathrm{I}[h(\mathbf{x}) \neq y]\mathrm{I}[h'(\mathbf{x}) \neq y], \tag{20}$$

*where $\mathrm{I}[\cdot]$ is an indicator function with $\mathrm{I}[\text{True}] = 1$ and $\mathrm{I}[\text{False}] = 0$. The $\epsilon$ in Eq. (11) is a value larger than the maximum $\beta$-Divergence in $\Lambda_S$,*

$$\epsilon \geq \max_{i,j\in[N_S],i\neq j,t\in[0,+\infty)} 2^{\frac{q-1}{q}\mathrm{RD}_q(\mathcal{S}_i(t)\|\mathcal{S}_j(t))}, \tag{21}$$

*where*

$$\mathrm{RD}_q(\mathcal{S}_i(t)\|\mathcal{S}_j(t)) = \frac{q(\mu_{j,t} - \mu_{i,t})^2}{2(1-q)\sigma_{i,t}^2 + 2\sigma_{j,t}^2} + \frac{\ln\frac{\sqrt{(1-q)\sigma_{i,t}^2 + \sigma_{j,t}^2}}{\sigma_{i,t}^{1-q}\sigma_{j,t}^q}}{1-q} \tag{22}$$

**Proof.** According to Theorem 3 of Germain et al. (2016), if $\mathcal{H}$ is a hypothesis space, and $\mathcal{S}, \mathcal{T}$ respectively are the source and target domains. For all $\rho$ in $\mathcal{H}$,

$$R_{\mathcal{T}}[\rho] \leq \frac{1}{2}\mathrm{d}_{\mathcal{T}}(\rho) + \beta_q(\mathcal{T}\|\mathcal{S})\cdot[\mathrm{e}_{\mathcal{S}}(\rho)]^{1-\frac{1}{q}} + \eta_{\mathcal{T}\backslash\mathcal{S}}, \tag{23}$$

where $\eta_{\mathcal{T}\backslash\mathcal{S}}$ denotes the distribution of $(\mathbf{x},y) \sim \mathcal{T}$ conditional to $(\mathbf{x},y) \in \mathrm{SUPP}(\mathcal{S})$. But because it is hardly conceivable to estimate the joint error $\mathrm{e}_{\mathcal{T}\backslash\mathcal{S}}(\rho)$ without making extra assumptions, Germain et al. (2016) defines the worst risk for this unknown area,

$$\eta_{\mathcal{T}\backslash\mathcal{S}} = \Pr_{(\mathbf{x},y)\sim\mathcal{T}}[(\mathbf{x},y) \notin \mathrm{SUPP}(\mathcal{S})]\sup_{h\in\mathcal{H}} R_{\mathcal{T}\backslash\mathcal{S}}[h]. \tag{24}$$

In Theorem 2.5, all domains from the convex hull $\Lambda_S$ have the same value range, in other words, their supports are continuous and fully overlapped. In this case, $\Pr_{(\mathbf{x},y)\sim\mathcal{T}}[(\mathbf{x},y) \notin \mathrm{SUPP}(\mathcal{S})] = 0$, i.e., $\eta_{\mathcal{T}\backslash\mathcal{S}} = 0$.

With Eq. (23), if the target domain $\mathcal{T}$ is assumed as an unseen domain $\mathcal{D}_{\mathrm{U}}$ from the convex hull $\Lambda_S$, and we select its closest element $\mathcal{D}_{\bar{\mathrm{U}}} = \arg\min_{\pi_1,\ldots,\pi_{N_S}} \beta_q(\mathcal{D}_{\bar{\mathrm{U}}}\|\sum_{i=1}^{N_S}\pi_i\mathcal{S}_i)$ and regard it as the source domain, we can derive Eq. (23) into

$$R_{\mathcal{D}_{\mathrm{U}}}[\rho] \leq \frac{1}{2}\mathrm{d}_{\mathcal{D}_{\mathrm{U}}}(\rho) + \beta_q(\mathcal{D}_{\mathrm{U}}\|\mathcal{D}_{\bar{\mathrm{U}}})\cdot\left[\mathrm{e}_{\mathcal{D}_{\bar{\mathrm{U}}}}(\rho)\right]^{1-\frac{1}{q}} + 0. \tag{25}$$

Then according to Lemma 2.4, as both $\mathcal{D}_{\mathrm{U}}$ and $\mathcal{D}_{\bar{\mathrm{U}}}$ are from the convex hull $\Lambda_S$, $\beta_q(\mathcal{D}_{\mathrm{U}}\|\mathcal{D}_{\bar{\mathrm{U}}}) \leq \epsilon$. As for acquiring Eq. (13), we only need to substitute the time series domains in the form of random variable distributions into the Rényi Divergence.

$$\square$$

***Theorem** 2.6. Suppose there are $M_{\mathcal{D}}$ samples (observations) available for a non-stationary time-series domain $\mathcal{D}_{\mathbf{x}}$, and each sample $\mathbf{x}_i = \{x_{i,0}, \ldots, x_{i,t}, \ldots\}$ is characterized by its deterministic function, i.e., $\mathbf{x}_i(t) = x_{i,t} = \mathbf{x}_i(t), i \in [1, M_{\mathcal{D}}]$. If we apply Hilbert Transformation $\mathrm{HT}(\mathbf{x}(t)) = \hat{\mathbf{x}}(t) = \int_{-\infty}^{\infty}\mathbf{x}(\tau)\frac{1}{\pi(t-\tau)}d\tau$ to augment these time-series samples, the non-stationary statistics of augmented samples are different from the original ones,*

$$\Pr_{\mathbf{x}\sim\widehat{\mathcal{D}}_{\mathbf{x}}}(\mathbf{x})(t) \neq \Pr_{\mathbf{x}\sim\mathcal{D}_{\mathbf{x}}}(\mathbf{x})(t). \tag{26}$$

**Proof.** According to Definition 2.1, the statistics of the non-stationary time-series domain consist of non-stationary mean and variance. To prove Theorem 2.6, we only need to prove that the mean of the time-series domain changes after applying Hilbert Transformation (HT). HT can only be conducted on deterministic signals, thus we use the empirical statistics of $M_{\mathcal{D}}$ samples to approximate the real statistics,

$$\mathbb{E}_{\mathbf{x}\sim\widehat{\mathcal{D}}_{\mathbf{x}}}(\mathbf{x})(t) = \sum_{i=1}^{M_{\mathcal{D}}}\widehat{\mathbf{x}}_i(t) = \widehat{\mu}_t, \quad \mathbb{E}_{\mathbf{x}\sim\mathcal{D}_{\mathbf{x}}}(\mathbf{x})(t) = \sum_{i=1}^{M_{\mathcal{D}}}\mathbf{x}_i(t) = \mu_t. \tag{27}$$

According to the standard definition of HT (King, 2009) and the linear property of integral operation, we have

$$\mathbb{E}_{\mathbf{x} \sim \widehat{\mathcal{D}}_{\mathbf{x}}}(\mathbf{x})(t) = \sum_{i=1}^{M_{\mathcal{D}}} \widehat{\mathbf{x}}_i(t) = \sum_{i=1}^{M_{\mathcal{D}}} \int_{-\infty}^{\infty} \mathbf{x}_i(\tau) \frac{1}{\pi(t-\tau)} d\tau = \int_{-\infty}^{\infty} \sum_{i=1}^{M_{\mathcal{D}}} \left[ \mathbf{x}_i(\tau) \frac{1}{\pi(t-\tau)} d\tau \right]$$
$$= \frac{1}{\pi} \int_{-\infty}^{\infty} \frac{\mu_\tau}{t-\tau} d\tau. \tag{28}$$

To interpret Eq. (28), we can assume there is a new signal $\mathbf{s} = \{\mu_0, ..., \mu_t, ...\}$ with the deterministic function $\mu_t = \mathrm{u}(t)$, and we next apply proof by contradiction for the following proof. Suppose the non-stationary statistics of the original and HT-transformed samples are identical, i.e., $\mathbb{E}_{\mathbf{x} \sim \widehat{\mathcal{D}}_{\mathbf{x}}}(\mathbf{x})(t) = \mathbb{E}_{\mathbf{x} \sim \mathcal{D}_{\mathbf{x}}}(\mathbf{x})(t)$, we can derive the following formula,

$$\frac{1}{\pi} \int_{-\infty}^{\infty} \frac{\mathrm{u}(\tau)}{t-\tau} d\tau = \mathrm{u}(t), \tag{29}$$

which indicates that the HT-transformed $\widehat{\mathbf{s}}$ is identical to the original $\mathbf{s}$. HT has a property called Orthogonality (King, 2009): if $\mathbf{x}(t)$ is a real-valued energy signal, then $\mathbf{x}(t)$ and its HT-transformed signal $\widehat{\mathbf{x}}(t)$ are orthogonal, i.e.,

$$\int_{-\infty}^{\infty} \mathbf{x}(t)\widehat{\mathbf{x}}(t)dt = 0. \tag{30}$$

To prove the property of Orthogonality, we need to use Plancherel's Formula,

**Theorem A.1** (Plancherel's Formula (Lang & Lang, 1985)). *Suppose that $u, v \in L^1(\mathbb{R}) \cap L^2(\mathbb{R})$, then*

$$\int_{-\infty}^{\infty} u(t)\overline{v(t)}dt = \frac{1}{2\pi} \int_{-\infty}^{\infty} \mathcal{F}u(\omega)\overline{\mathcal{F}v(\omega)}d\omega, \tag{31}$$

*where $L^1(\cdot), L^2(\cdot)$ denote the $L^p$ spaces with $p = 1, p = 2$ respectively, $\mathbb{R}$ represents the real-valued space, and $\mathcal{F}$ denotes the Plancherel transformation.*

With Plancherel's Formula, we can prove the property of Orthogonality as follows,

$$\int_{-\infty}^{\infty} \mathbf{x}(t)\widehat{\mathbf{x}}(t)dt = \frac{1}{2\pi} \int_{-\infty}^{\infty} \mathcal{F}(\omega)(-i\,\mathrm{sgn}(\omega)\mathcal{F}(\omega))^* d\omega$$
$$= \frac{i}{2\pi} \int_{-\infty}^{\infty} \mathrm{sgn}(\omega)\mathcal{F}(\omega)\mathcal{F}^*(\omega)d\omega$$
$$= \frac{i}{2\pi} \int_{-\infty}^{\infty} \mathrm{sgn}(\omega)|\mathcal{F}(\omega)|^2 d\omega \tag{32}$$
$$= 0,$$

where $\mathrm{sgn}(\cdot)$ is a sign function. After proving the Orthogonality, we can use it with the condition of Eq. (29), i.e.,

$$\int_{-\infty}^{\infty} \mathrm{u}(t)\widehat{\mathrm{u}}(t)dt = \int_{-\infty}^{\infty} \mathrm{u}^2(t)dt = 0. \tag{33}$$

Eq. (33) holds true only if $\forall t \in [0, +\infty), \mathrm{u}(t) = 0$, which is contradict to our initial assumption that $\mu_t = \mathrm{u}(t)$ is not always zero in Definition 2.1. As a result, the assumption of $\widehat{\mu}_t = \mu_t$ is false. $\square$

## B  ADDITIONAL DETAILS ON PHASER

**Augmented Dickey Fuller (ADF) Test.** This is a statistical tool to assess the non-stationarity of a given time-series signal. This test operates under a null hypothesis $\mathbb{H}_0$ where the signal has a *unit-root*. The existence of *unit-root* is a guarantee that the signal is non-stationary (Said & Dickey, 1984). To reject $\mathbb{H}_0$, the statistic value of the ADF test should be less than the critical values associated with a

significance level of $0.05$ (denoted by $p$, the probability of observing such a test statistic under the null hypothesis). Throughout the paper, for multivariate time series, the average ADF statistics across all variates are reported. Besides, since this is a statistical tool to evaluate non-stationarity for each instance of time-series data, we provide an average of this number across a dataset to give the reader a view of the degree of non-stationarity.

**Phase Augmentation.** In this work, we are particularly interested in learning representations robust to temporal distribution shifts. Incorporating a phase shift in a signal is a less-studied augmentation technique. One of the main challenges is that real-world signals are not composed of a single frequency component and accurately estimating and controlling the shifting of the phase while retaining the magnitude spectrum of a signal is difficult. To solve this, we leverage the analytic transformation of a signal using the Hilbert Transform. The key advantages of this technique are maintaining global temporal dependencies and magnitude spectrum, no exploration of design parameters and being extendible to non-stationary and periodic time series.

Lets walk through a simple example for a signal, $\mathbf{x}(t) = 2cos(w_0 t)$ which can be written in the polar coordinates as $\mathbf{x}(t) = e^{iw_0 t} + e^{-iw_0 t}$. Applying the HT conditions from Equation 4, $\text{HT}(\mathbf{x}(t)) = 2sin(w_0 t)$. Essentially, HT shifts the signal by $\pi/2$ radians. We conduct this instance-level augmentation for each variate of the time series input. The aim is to diversify the phase representation. We use the *scipy* (Virtanen et al., 2020) library to implement this augmentation.

**STFT Specifications.** Non-stationary signals contain time-varying spectral properties. We use STFT to capture these magnitude and phase responses in both time and frequency domains. There are three main arguments to compute STFT - length of each segment (characterized by the window size and the ratio for overlap), the number of frequency bins, and the sampling rate. We use the scipy library to implement this operation and use a $k < 1$ as a multiplier to the length of the window $W$ to give the segment length as $k \times W$ with no overlap between segments. The complete list of STFT specifications is given in Table 6. We also demonstrate a sensitivity analysis concerning the number of frequency bins and the segment length in Figure 7.

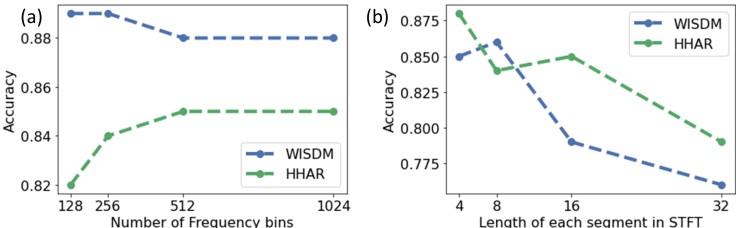

Figure 7: Illustration of the sensitivity of performance to the design choices of STFT by varying a) the number of frequency bins with a fixed segment length of 4 and b) by varying the segment lengths with a 1024 frequency bins.

Table 6: Arguments for STFT computation

| Dataset | Sampling Rate | Sequence Length | STFT segment length | Number of frequency bins |
|---|---|---|---|---|
| WISDM | 20 Hz | 128 | 4 | 1024 |
| HHAR | 100 Hz | 128 | 4 | 1024 |
| UCIHAR | 50 Hz | 128 | 4 | 1024 |
| SSC | 100 Hz | 3000 | 16 | 1024 |
| GR | 200 Hz | 200 | 4 | 1024 |

*Note:* It is tempting to use an empirical mode transformation and then apply a Hilbert-Huang transformation to obtain an instantaneous phase and amplitude response in the case of non-stationary signals. It absolves us from a finite time-frequency resolution for the STFT spectra. However, our initial results indicate a high dependence on the choice of the number of intrinsic mode functions (Huang, 2014) for signal decomposition. Hence, for a generalizable approach, we choose STFT as the tool for the time-frequency spectrum.

**Backbones for Temporal Encoder.** The choice of temporal encoder, $F_{\text{Tem}}$, is not central to our design. Table 7 demonstrates the performance of PhASER under the identical settings for four cross-

person settings using WISDM datasets using different backbones for $F_{\text{Tem}}$. For the convolution-based self-attention (second row in Table 7) we use three encoders to compute query ($W_q$), key ($W_k$), and value($V$) matrices for $\mathbf{r}_{\text{Dep}}$ following the guidelines from Vaswani et al. (2017). Then we compute self-attention as, $A = softmax\left(\frac{QK^T}{\sqrt{d_k}}\right)V$, where $d_k$ is the temporal dimension of $\mathbf{r}_{\text{Dep}}$. Subsequently, we use $\hat{\mathbf{r}}_{\text{Dep}} = \mathbf{r}_{\text{Dep}} + A$, as the input to $F_{\text{Tem}}$. For more details on the convolution and transformer backbones refer to Section D.3.

Table 7: Results for 4 different cross-person settings for WISDM dataset.

| Backbones for $F_{\text{Tem}}$ | 1 | 2 | 3 | 4 |
|---|---|---|---|---|
| 2D Convolution based | 0.86 | 0.85 | 0.86 | 0.84 |
| 2D Convolution based with self-attention | 0.88 | 0.83 | 0.84 | 0.81 |
| Transformer | 0.87 | 0.84 | 0.87 | 0.84 |

## C  DATASET DETAILS

Past works (Gagnon-Audet et al., 2022; Ragab et al., 2023a) have shown that the datasets used in our work suffer from a distribution shift across users and also within the same user temporally. This makes them suitable for evaluating the efficacy of our framework. In this section, we provide more details on the datasets. Table 8 summarizes the average ADF statistics of the datasets along with their variates and their number of classes and domains.

Table 8: Summary of the dataset attributes. Higher value of ADF stat indicates greater non-stationarity within a signal.

| Category | Dataset | Representative ADF-Statistic (mean across all variates) | Variates | Domains | Classes |
|---|---|---|---|---|---|
| Human Activity recognition | UCIHAR | -2.58 (0.044) | 9 | 31 | 6 |
| Human Activity recognition | HHAR | -1.74 (0.062) | 3 | 9 | 6 |
| Human Activity recognition | WISDM | -0.78 (0.051) | 3 | 36 | 6 |
| Gesture Recognition | EMG | -33.14 (0.011) | 8 | 36 | 6 |
| Sleep Stage Classification | EEG | -3.7 (0.047) | 1 | 20 | 5 |

**WISDM** (Kwapisz et al., 2011): It originally consists of 51 subjects performing 18 activities but we follow the ADATime (Ragab et al., 2023a) suite to utilize 36 subjects comprising of 6 activity classes given as walking, climbing upstairs, climbing downstairs, sitting, standing, and lying down. The dataset consists of 3-axis accelerometer measurements sampled at 20 Hz to predict the activity of each participant for a segment of 128-time steps. According to Ragab et al. (2023a), this is the most challenging dataset suffering from the highest degree of class imbalance.

**HHAR** (Stisen et al., 2015): To remain consistent with the existing AdaTime benchmark we leverage the Samsung Galaxy recordings of this dataset from 9 participants from a 3-axis accelerometer sampled at 100 Hz. The 6 activity classes, in this case, are - biking, sitting, standing, walking, climbing up the stairs, and climbing down the stairs.

**UCIHAR** (Bulbul et al., 2018): This dataset is collected from 30 participants using 9-axis inertial motion unit using a waist-mounted cellular device sampled at 50 Hz. The six activity classes are the same as WISDM dataset.

**SSC** (Goldberger et al., 2000): This is a single channel EEG dataset collected from 20 subjects to classify five sleep stages - wake, non-rapid eye movement stages - N1, N2, N3, and rapid-eye-movement.

**GR** (Lobov et al., 2018): For surface-EMG based gesture recognition we follow Lu et al. (2023)'s preprocessing and use an 8-channel data recorded from 36 participants for six types of gestures sampled at 200 Hz. Note, that this is the least stationary dataset (see Table 8, yet `PhASER` performs as well as or better than the stat-of-the-art techniques as shown in Table 4 in the main paper.

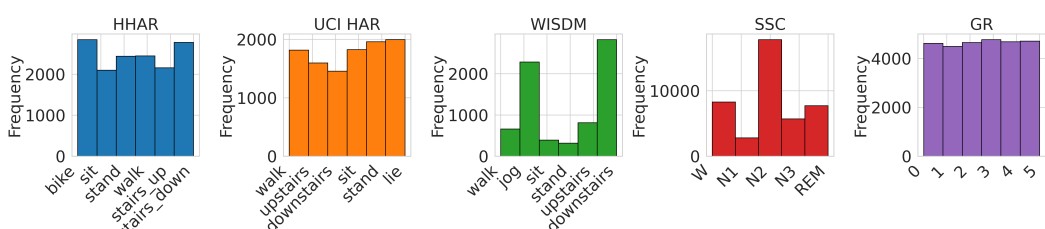

Figure 8: Class Distributions of the datasets used for evaluation.

## D  IMPLEMENTATION DETAILS

All experiments are performed on an Ubuntu OS server equipped with NVIDIA TITAN RTX GPU cards using PyTorch framework. Every experiment is carried out with 3 different seeds (2711, 2712, 2713). During model training, we use Adam optimizer (Kingma et al., 2020) with a learning rate from 1e-5 to 1e-3 and maximum number of epochs is set to 150 based on the suitability of each setting. We tune these optimization-related hyperparameters for each setting and save the best model checkpoint based on early exit based on the minimum value of the loss function achieved on the validation set.

### D.1  DATASET CONFIGURATION

There is no standard benchmarking for domain generalization for time-series where the domain labels and target samples are inaccessible. We leverage past works of Ragab et al. (2023a); Lu et al. (2023) for preprocessing steps. For each dataset, we use a cross-person setting in four scenarios. The details of the target domains chosen in each scenario are given in Table 9, the rest are used as source domains. Note for GR we use the same splits as Lu et al. (2023). Our method is not influenced by domain labels as we do not require them for our optimization.

Table 9: Target domain splits for 4 scenarios of each dataset.

| Target Domains | Scenario 1 | Scenario 2 | Scenario 3 | Scenario 4 |
|---|---|---|---|---|
| WISDM | 0-9 | 10-17 | 18-27 | 28-35 |
| HHAR | 0,1 | 2,3 | 4,5 | 6-8 |
| UCIHAR | 0-7 | 8-15 | 16-23 | 24-29 |
| GR | 0-8 | 9-17 | 18-26 | 27-35 |
| SSC | 0-5 | 5-9 | 10-14 | 15-20 |

Figure D.1 illustrates the class distribution for each dataset. Only the WISDM and Sleep Stage Classification (SSC) datasets exhibit notable imbalances among certain classes. To validate the consistency of our conclusions, we compare the Area Under the Curve (AUC) with the adopted accuracy metric in Figure D.1. Generally, past works (Lu et al., 2023; Gagnon-Audet et al., 2022), utilizing these datasets have adopted accuracy as the primary performance metric, and we follow the same approach.

### D.2  BASELINE METHODS

**General Domain Generalization Methods.** For all the standard domain generalization baselines we use conv2D layers for feature transformation of multivariate time series. It is worth mentioning that DANN is actually a domain adaptation study, which requires access to certain unlabeled target domain data. For cross-person generalization, the source domain consists of data from multiple people, in which we divide the source domain data into two parts with equal size and view one of them as the target domain to leverage DANN for domain-invariant training. As for one-person-to-another cases, we randomly sample a small number of unlabeled instances from each target person and merge them into the target set that is needed for running DANN.

**BCResNet.** This is a competitive benchmark for several audio-scene recognition challenges and demonstrates many useful techniques for domain generalization. BCResNet originally required

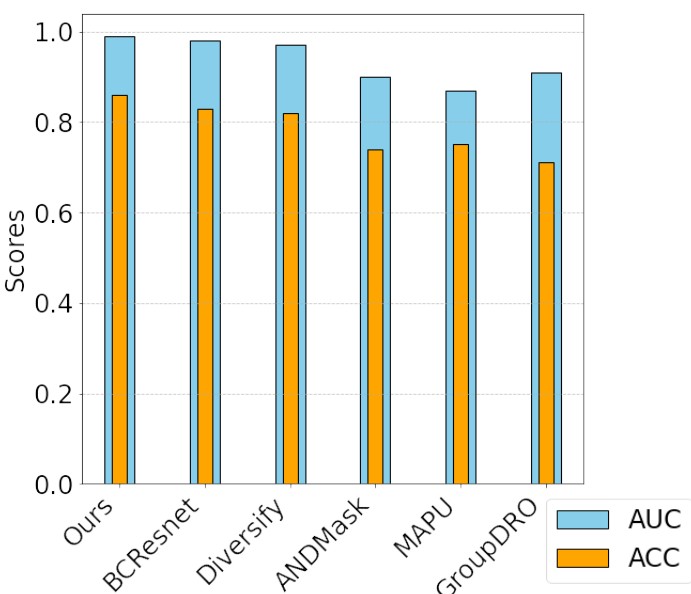

Figure 9: Illustration of additional performance metric, Area Under the ROC Curve (AUC), along with Accuracy—for Scenario 1 of the WISDM dataset, for the top-performing baselines. These metrics demonstrate consistency and justify our choice of accuracy as the primary evaluation metric.

mel-frequency-cepstral-coefficients but it is not suitable for time-series, hence, we use standard STFT of the multivariate-time series as input in this case.

**Non-Stationary Transformer and Koopa.** These are forecasting baselines that particularly address non-stationarity in short-term time sequences, Non-stationary transformer (NSTrans) (Liu et al., 2022) and Koopa (Liu et al., 2024). To adapt it to our setting we use the encoder part of NSTrans followed by a classification head composed of fully connected layers. We simply average the encoder's output from all time steps and feed it to this classifier head.

**Ours+RevIN.** Further, we demonstrate that statistical techniques like Reversible Instance Normalization (RevIN) (Kim et al., 2021c) may be used as a plug-and-play module with our framework. One limitation of using RevIN is that the input and output dimensions of this module must have the same dimensions to de-normalize the instance in the feature space. This may limit the usability of the module, however, we find that applying this module around the fusion encoder specifying the same number of input and output channels in the 2D convolution layer is suitable. We do not observe any significant benefit of incorporating this module from the experiments, however, if an application can specifically benefit from such RevIN, `PhASER` framework can support it.

**Diversify.** The goal of this design is to characterize the latent domains and use a proxy-training schema to assign pseudo-domain labels to the samples to learn generalizable representations. It is an end-to-end version of the adaptive RNN (Du et al., 2021) method which also proposes to identify sub-domains within a domain for generalization. It is interesting to note that for time-series generalizable representation viewing the non-stationarity or intra-domain shifts is crucial. Both diversify and `PhASER` address this problem from completely different approaches and demonstrate improvement over other standard methods or even domain adaptation methods that have the advantage of accessing samples from unseen distributions. While diversify aims to characterize latent distributions and uses a parametric setting, `PhASER` forces the model to learn domain-invariant features by anchoring the design to the phase which is intricately tied to non-stationarity. It also highlights that time-series domain generalization is a unique problem (compared to the more popular visual domain) and dedicated frameworks need to be designed in this case.

**MAPU.** MAPU is the state-of-the-art source-free domain adaptation study for time series, thus, in fact, it does not apply to the time-series domain generalizable learning problem. However, we still view it as an effective approach that can address distribution shifts and achieve domain-invariant

learning. In our implementation, in addition to the source domain data, we still provide MAPU with the unlabeled target domain data for both cross-person generalization and one-person-to-another cases. The training procedure is identical to the default MAPU design, which is to pre-train the model on labeled source domain data and then conduct the training on unlabeled target domain data.

**Chronos.** Large foundation models are a sought-after approach in many domains and Chronos is one such most recent candidate for time-series. It is trained on 42 datasets and presents impressive zero-shot and few-shot abilities. Although it is largely targeted as a forecasting tool, the authors indicate its universal representation ability for a variety of tasks. Four variants of Chronos model checkpoints are available ranging from 20M to 70M parameters and embedding sizes from 256 to 1024. Based on pilot testing with scenario 1 on WISDM dataset (accuracies with a 1M parameter downstream model for the three variants: tiny-0.65, base-0.41, large-0.36), we find that the smallest version of the model, Chronos-tiny best suits our conservative dataset sizes for downstream fine-tuning. We use a few layers of 2D convolution layers with max-pooling to reduce the feature size which is dependent of the length of the sequence and then flatten and input to fully-connected layers as our downstream model.

*Note:* A few works (Jin et al., 2024; Liu et al., 2023a) use large language models directly to analyze raw time-series despite the obvious modality gap and can report comparable performance. However, our preliminary testing with ChatGPT (Radford et al., 2019) with in-context-learning by prompting similar to Jin et. al (Jin et al., 2024) using the HHAR dataset does not provide satisfactory results and we do not pursue that direction. Instead, we use a domain-specific large foundation model like Chronos as a fair baseline.

Table 10: Complete set of results from three trials on each baseline for WISDM cross-person generalization setting.

| Baselines | Scenario 1 | | Scenario 2 | | Scenario 3 | | Scenario 4 | |
|---|---|---|---|---|---|---|---|---|
| | Mean | Std | Mean | Std | Mean | Std | Mean | Std |
| ERM | 0.57 | 0.02 | 0.50 | 0.02 | 0.51 | 0.02 | 0.55 | 0.02 |
| GroupDRO | 0.71 | 0.06 | 0.67 | 0.06 | 0.60 | 0.07 | 0.67 | 0.04 |
| DANN | 0.71 | 0.02 | 0.65 | 0.01 | 0.65 | 0.06 | 0.70 | 0.03 |
| RSC | 0.69 | 0.05 | 0.71 | 0.07 | 0.64 | 0.10 | 0.61 | 0.11 |
| ANDMask | 0.74 | 0.01 | 0.73 | 0.03 | 0.69 | 0.06 | 0.69 | 0.03 |
| BCResNet | 0.83 | 0.00 | 0.79 | 0.04 | 0.75 | 0.04 | 0.78 | 0.04 |
| NSTrans | 0.43 | 0.02 | 0.40 | 0.01 | 0.37 | 0.02 | 0.37 | 0.03 |
| Koopa | 0.63 | 0.02 | 0.61 | 0.04 | 0.72 | 0.03 | 0.57 | 0.01 |
| MAPU | 0.75 | 0.02 | 0.69 | 0.04 | 0.79 | 0.06 | 0.79 | 0.03 |
| Diversify | 0.82 | 0.01 | 0.82 | 0.01 | 0.84 | 0.01 | 0.81 | 0.01 |
| Chronos | 0.71 | 0.01 | 0.67 | 0.01 | 0.65 | 0.01 | 0.62 | 0.01 |
| Ours + RevIN* | 0.86 | 0.01 | 0.85 | 0.01 | 0.84 | 0 | 0.84 | 0.03 |
| Ours | 0.86 | 0.01 | 0.85 | 0.01 | 0.85 | 0.01 | 0.82 | 0.02 |

### D.3 IMPLEMENTATION DETAILS OF PHASER

The magnitude and phase encoders, $F_{\mathrm{Mag}}$ and $F_{\mathrm{Pha}}$ are implemented using 2D convolution layers with the number of input channels equal to the variates, $V$, and the out channels as $2c$ with $(5 \times 5)$ kernels. $c$ is a hyperparameter used to conveniently control the size of the overall network. For all HAR and GR models we adopt $c$ as 1 and for SSC $c$ is 4. For more specific details please refer to our code. The sub-spectral feature normalization uses a group number of 3 and follows Equation 2.3 for operation. This is inspired by Chang et. al (Chang et al., 2021) subspectral normalization for audio applications with a frequency spectrum input. The key idea is to conduct sub-band normalization (across a fixed set of frequency bins along time and examples for each channel). We find merit in using this technique for domain generalizable applications, as it can help overcome the low-frequency drifts arising due to device differences (for eg. DC drifts in various sensors). One implementation-specific modification we carried out to ensure a generalizable framework is that if the number of sub-bands is not divisible by the total number of features then we choose to apply the remainder bands with batch-normalization. The output from the respective encoders is then fused along the channel/variate axis by multiplying with 2D convolution kernels to provide a new feature map which is the input to our phase-driven residual network. The $F_{\mathrm{Fus}}$ similarly is implemented using 2D convolution layers with the number of input channels as $4c$ and output channels to be $2c$.

Table 11: Complete set of results from three trials on each baseline for HHAR cross-person generalization setting.

| Baselines | Scenario 1 | | Scenario 2 | | Scenario 3 | | Scenario 4 | |
|---|---|---|---|---|---|---|---|---|
| | Mean | Std | Mean | Std | Mean | Std | Mean | Std |
| ERM | 0.49 | 0.05 | 0.46 | 0.01 | 0.45 | 0.02 | 0.47 | 0.03 |
| GroupDRO | 0.60 | 0.01 | 0.53 | 0.02 | 0.59 | 0.02 | 0.64 | 0.03 |
| DANN | 0.66 | 0.01 | 0.71 | 0.01 | 0.67 | 0.09 | 0.69 | 0.03 |
| RSC | 0.52 | 0.05 | 0.49 | 0.04 | 0.44 | 0.03 | 0.47 | 0.03 |
| ANDMask | 0.63 | 0.02 | 0.64 | 0.06 | 0.66 | 0.11 | 0.69 | 0.05 |
| BCResNet | 0.66 | 0.05 | 0.70 | 0.06 | 0.75 | 0.04 | 0.68 | 0.04 |
| NSTrans | 0.21 | 0.02 | 0.22 | 0.03 | 0.27 | 0.04 | 0.28 | 0.02 |
| Koopa | 0.72 | 0.04 | 0.63 | 0.03 | 0.72 | 0.05 | 0.69 | 0.02 |
| MAPU | 0.73 | 0.02 | 0.72 | 0.03 | 0.81 | 0.01 | 0.78 | 0.03 |
| Diversify | 0.82 | 0.01 | 0.76 | 0.01 | 0.82 | 0.01 | 0.68 | 0.01 |
| Chronos | 0.73 | 0.04 | 0.75 | 0.03 | 0.73 | 0.01 | 0.66 | 0.12 |
| Ours + RevIN* | 0.82 | 0.05 | 0.82 | 0.02 | 0.92 | 0.04 | 0.85 | 0.03 |
| Ours | 0.83 | 0.02 | 0.83 | 0.02 | 0.94 | 0.03 | 0.88 | 0.02 |

Table 12: Complete set of results from three trials on each baseline for UCIHAR cross-person generalization setting.

| Baselines | Scenario 1 | | Scenario 2 | | Scenario 3 | | Scenario 4 | |
|---|---|---|---|---|---|---|---|---|
| | Mean | Std | Mean | Std | Mean | Std | Mean | Std |
| ERM | 0.72 | 0.09 | 0.64 | 0.05 | 0.70 | 0.01 | 0.72 | 0.03 |
| GroupDRO | 0.91 | 0.02 | 0.84 | 0.01 | 0.89 | 0.04 | 0.85 | 0.07 |
| DANN | 0.84 | 0.02 | 0.79 | 0.01 | 0.81 | 0.02 | 0.86 | 0.03 |
| RSC | 0.82 | 0.13 | 0.73 | 0.07 | 0.74 | 0.03 | 0.81 | 0.06 |
| ANDMask | 0.86 | 0.08 | 0.80 | 0.06 | 0.76 | 0.13 | 0.78 | 0.09 |
| BCResNet | 0.81 | 0.02 | 0.77 | 0.02 | 0.78 | 0.02 | 0.83 | 0.02 |
| NSTrans | 0.35 | 0.02 | 0.35 | 0.01 | 0.51 | 0.02 | 0.47 | 0.01 |
| Koopa | 0.81 | 0.02 | 0.72 | 0.05 | 0.81 | 0.06 | 0.77 | 0.03 |
| MAPU | 0.85 | 0.03 | 0.80 | 0.01 | 0.85 | 0.02 | 0.82 | 0.03 |
| Diversify | 0.89 | 0.03 | 0.84 | 0.04 | 0.93 | 0.02 | 0.90 | 0.02 |
| Chronos | 0.56 | 0.05 | 0.57 | 0.01 | 0.50 | 0.02 | 0.82 | 0.13 |
| Ours + RevIN* | 0.96 | 0.01 | 0.90 | 0.01 | 0.93 | 0.03 | 0.97 | 0.01 |
| Ours | 0.96 | 0.01 | 0.91 | 0.01 | 0.95 | 0 | 0.97 | 0.01 |

Subsequently for the depth-wise encoder, $F_{\text{Dep}}$, we use 2D convolution layers with batch normalization and SiLU (Elfwing et al., 2018) activation function. This style of architecture is closely adapted from the basic building blocks in BCResNet (Kim et al., 2021a). After average pooling the $F_{\text{Tem}}$ can assume any backbone as per the requirements of the application. As demonstrated previously in Section B, the choice of backbone is not central to our design here. We find that some applications(like WISDM and GR) benefit from attention-based temporal encoding more than others. For the attention-based version of $F_{\text{Tem}}$ we used a multi-headed attention based on a transformer encoder (Vaswani et al., 2017). Regarding positional encoding, we used a simple sinusoid-based encoding and added it to the sequence representation $\mathbf{r}_{\text{Dep}}$. However, arriving at the best positional encoding for numerical time-series data is an active area of research (Kazemi et al., 2019; Tang et al., 2023; Mohapatra et al., 2023) given its uniqueness compared to typical natural language inputs and further optimizations can be carried out. For the the convolution-based $F_{\text{Tem}}$ we simply use a kernel of size $(1 \times 3)$ in a 2D convolution layer to conduct temporal convolutions.

For the classification head, $g_{\text{Cls}}$, we apply 2D convolution layers to have the number of output channels equal to the number of classes in an application, followed by softmax operation. Interestingly, if the choice of $F_{\text{Tem}}$ remains convolutional the entire network can be implemented in a purely convolutional form allowing applicability to real-time problems. The model sizes across the different datasets range from 40k-100k trainable parameters (based on the number of variates, temporal encoding etc.) which is modest and can be further tuned for resource-constrained applications by adjusting the $c$ parameter.

Table 13: Complete set of results from three trials on each baseline for SSC cross-person generalization setting.

| Baselines | Scenario 1 | | Scenario 2 | | Scenario 3 | | Scenario 4 | |
|---|---|---|---|---|---|---|---|---|
| | Mean | Std | Mean | Std | Mean | Std | Mean | Std |
| ERM | 0.50 | 0.05 | 0.46 | 0.04 | 0.49 | 0.02 | 0.45 | 0.03 |
| GroupDRO | 0.57 | 0.07 | 0.56 | 0.03 | 0.55 | 0.05 | 0.59 | 0.06 |
| DANN | 0.64 | 0.02 | 0.63 | 0.02 | 0.69 | 0.03 | 0.63 | 0.04 |
| RSC | 0.50 | 0.09 | 0.48 | 0.02 | 0.52 | 0.07 | 0.46 | 0.01 |
| ANDMask | 0.55 | 0.10 | 0.50 | 0.09 | 0.54 | 0.07 | 0.57 | 0.08 |
| BCResNet | 0.79 | 0 | 0.82 | 0.01 | 0.79 | 0.01 | 0.81 | 0 |
| NSTrans | 0.43 | 0.02 | 0.37 | 0.04 | 0.42 | 0.06 | 0.35 | 0.03 |
| Koopa | 0.58 | 0.02 | 0.62 | 0.01 | 0.53 | 0.04 | 0.49 | 0.06 |
| MAPU | 0.69 | 0.01 | 0.68 | 0.01 | 0.65 | 0.03 | 0.69 | 0.02 |
| Diversify | 0.73 | 0.03 | 0.76 | 0.02 | 0.68 | 0.05 | 0.77 | 0.02 |
| Chronos | 0.53 | 0.04 | 0.47 | 0.04 | 0.47 | 0.01 | 0.57 | 0.03 |
| Ours + RevIN* | 0.82 | 0.01 | 0.79 | 0.02 | 0.78 | 0.01 | 0.81 | 0.01 |
| Ours | 0.85 | 0.01 | 0.80 | 0.01 | 0.79 | 0.01 | 0.83 | 0.01 |

Table 14: Complete set of results from three trials on each baseline for GR cross-person generalization setting.

| Baselines | Scenario 1 | | Scenario 2 | | Scenario 3 | | Scenario 4 | |
|---|---|---|---|---|---|---|---|---|
| | Mean | Std | Mean | Std | Mean | Std | Mean | Std |
| ERM | 0.45 | 0.02 | 0.58 | 0.03 | 0.57 | 0.03 | 0.54 | 0.04 |
| GroupDRO | 0.53 | 0.08 | 0.36 | 0.11 | 0.59 | 0.05 | 0.45 | 0.13 |
| DANN | 0.60 | 0.01 | 0.66 | 0.04 | 0.65 | 0.02 | 0.64 | 0.03 |
| RSC | 0.50 | 0.10 | 0.66 | 0.05 | 0.64 | 0.03 | 0.56 | 0.03 |
| ANDMask | 0.41 | 0.13 | 0.54 | 0.20 | 0.45 | 0.15 | 0.39 | 0.12 |
| BCResNet | 0.62 | 0.06 | 0.67 | 0.09 | 0.65 | 0.05 | 0.61 | 0.07 |
| NSTrans | 0.31 | 0.01 | 0.34 | 0.01 | 0.34 | 0.01 | 0.32 | 0.02 |
| Koopa | 0.47 | 0.03 | 0.54 | 0.02 | 0.60 | 0.05 | 0.70 | 0.06 |
| MAPU | 0.64 | 0.02 | 0.69 | 0.03 | 0.71 | 0.01 | 0.68 | 0.04 |
| Diversify | 0.69 | 0.01 | 0.80 | 0.01 | 0.76 | 0.02 | 0.76 | 0.01 |
| Chronos | 0.49 | 0.01 | 0.54 | 0.03 | 0.51 | 0.05 | 0.48 | 0.02 |
| Ours + RevIN* | 0.68 | 0.03 | 0.81 | 0.04 | 0.77 | 0.03 | 0.76 | 0.02 |
| Ours | 0.70 | 0.02 | 0.82 | 0.02 | 0.77 | 0.04 | 0.75 | 0.01 |

Table 15: Complete set of results from three trials on each baseline for HHAR one-person-to-another setting.

| Baselines | 0 | | 1 | | 2 | | 3 | | 4 | | 5 | | 6 | | 7 | | 8 | |
|---|---|---|---|---|---|---|---|---|---|---|---|---|---|---|---|---|---|---|
| | Mean | Std | Mean | Std | Mean | Std | Mean | Std | Mean | Std | Mean | Std | Mean | Std | Mean | Std | Mean | Std |
| ERM | 0.27 | 0.01 | 0.40 | 0.05 | 0.41 | 0.05 | 0.44 | 0.05 | 0.42 | 0.08 | 0.44 | 0.01 | 0.45 | 0.04 | 0.44 | 0.04 | 0.48 | 0.02 |
| GroupDRO | 0.33 | 0.02 | 0.53 | 0.02 | 0.38 | 0.05 | 0.48 | 0.04 | 0.47 | 0.04 | 0.51 | 0.08 | 0.47 | 0.03 | 0.48 | 0.02 | 0.49 | 0.05 |
| DANN | 0.32 | 0.03 | 0.44 | 0.05 | 0.42 | 0.03 | 0.45 | 0.06 | 0.42 | 0.03 | 0.48 | 0.04 | 0.49 | 0.02 | 0.45 | 0.05 | 0.51 | 0.01 |
| RSC | 0.27 | 0.03 | 0.45 | 0.06 | 0.38 | 0.05 | 0.45 | 0.09 | 0.40 | 0.08 | 0.47 | 0.02 | 0.50 | 0.06 | 0.44 | 0.08 | 0.53 | 0.01 |
| ANDMask | 0.34 | 0.06 | 0.50 | 0.03 | 0.37 | 0.04 | 0.43 | 0.05 | 0.46 | 0.04 | 0.51 | 0.07 | 0.46 | 0.03 | 0.47 | 0.02 | 0.52 | 0.03 |
| BCResNet | 0.28 | 0.03 | 0.48 | 0.08 | 0.32 | 0.04 | 0.47 | 0.03 | 0.42 | 0.06 | 0.52 | 0.05 | 0.44 | 0.02 | 0.45 | 0.02 | 0.49 | 0.06 |
| NSTrans | 0.20 | 0.01 | 0.22 | 0.02 | 0.17 | 0.02 | 0.20 | 0.01 | 0.21 | 0.01 | 0.22 | 0.01 | 0.26 | 0.07 | 0.17 | 0.05 | 0.20 | 0.01 |
| Koopa | 0.32 | 0.02 | 0.42 | 0.04 | 0.37 | 0.01 | 0.40 | 0.01 | 0.42 | 0.02 | 0.45 | 0.05 | 0.35 | 0.02 | 0.43 | 0.03 | 0.48 | 0.02 |
| MAPU | 0.39 | 0.05 | 0.57 | 0.05 | 0.35 | 0.06 | 0.52 | 0.03 | 0.49 | 0.04 | 0.54 | 0.02 | 0.49 | 0.01 | 0.50 | 0.06 | 0.52 | 0.04 |
| Diversify | 0.42 | 0.04 | 0.62 | 0.04 | 0.32 | 0.09 | 0.62 | 0.01 | 0.56 | 0.03 | 0.61 | 0.01 | 0.53 | 0.04 | 0.52 | 0.10 | 0.61 | 0.05 |
| Chronos | 0.32 | 0.03 | 0.23 | 0.05 | 0.26 | 0.04 | 0.25 | 0.03 | 0.27 | 0.09 | 0.23 | 0.08 | 0.24 | 0.06 | 0.21 | 0.04 | 0.24 | 0.05 |
| Ours + RevIN* | 0.48 | 0.02 | 0.66 | 0.08 | 0.57 | 0.05 | 0.65 | 0.03 | 0.61 | 0.04 | 0.64 | 0.05 | 0.65 | 0.06 | 0.64 | 0.01 | 0.63 | 0.03 |
| Ours | 0.53 | 0.04 | 0.70 | 0.03 | 0.63 | 0.01 | 0.66 | 0.03 | 0.64 | 0.06 | 0.67 | 0.01 | 0.65 | 0.03 | 0.67 | 0.04 | 0.62 | 0.02 |

## D.4 ABLATION DETAILS OF PhASER

For row 1 in Table 5, the modification to PhASER is straightforward by simply omitted the Hilbert transformation during data preprocessing. When the separate encoders are not used (rows 6 and 7 in Table 5), we only use $F_{\text{Mag}}$ and connect the output of the sub-feature normalization block directly to the $F_{\text{Dep}}$. When the residual is removed entirely (rows 5 and 6 in Table 5), we cannot broadcast the 1D input to 2D anymore so we take the mean across all the temporal indices of $F_{\text{Tem}}(\mathbf{r}_{\text{Dep}})$ and flatten it to input to fully connected layers. Based on the dataset we choose a few fully connected layers truncating to the number of classes finally.

## D.5 PHASE-DRIVEN NSTRANS

Non-stationary transformer, NSTrans (Liu et al., 2022), applies a destationarizing attention around the transformer block. Since it is typically used for forecasting tasks, it comprises of encoder and a decoder module. For adapting this model to classification we update the design to conduct normalization and denormalization around the encoder block. We use this modified version of NSTrans as the $F_{\text{Tem}}$ module in PhASER and observe significant improvement in performance as shown in Figure 4.

*Note:* The poor performance of the Nonstationary transformer can be attributed to two main reasons:

(1) Originally, the Nonstationary transformer was designed for forecasting time-series tasks and employs an encoder-decoder style architecture. To successfully apply the core module of the Nonstationary transformer (Liu et al., 2022), stationarization-destationarization, the input-output space needs to remain consistent. This consistency is naturally ensured in an encoder-decoder design. However, in our classification applications, we only utilize the encoder module. Although we maintain the input-output dimensions, the semantics of the latent space and input space are not the same. Hence, destationarization is not very successful.

(2) Nonstationary transformer inputs consist of raw time-series data with positional encoding. Given the fine-grained nature of current tasks, such an approach can be more data-hungry as they try to establish a relation (attention) among every time step. Therefore, it may not perform well on short-range classification tasks that focus on domain generalization. This indicates a limitation in its direct usage for optimizing a categorical objective function using only the encoder part with a classification head.

## D.6 COMPUTATIONAL ANALYSES

To assess the resource utilization of PhASER against other baselines, we offer two metrics - 1) Number of Multiply and Accumulate operations per sample (MACs) for approximate computational complexity at run-time and 2) Number of trainable parameters to determine the memory footprint. We compute these for the HHAR dataset in Table 16 (these metrics are dependent on input dimensions, hence different choices of dataset, sequence length, and modalities can yield different numbers).

Table 16: Model comparison based on MACs and number of trainable parameters.

| Model | MACs ($\times 10^6$) | Trainable Parameters ($\times 10^3$) |
|---|---|---|
| ERM | 19.5 | 98.1 |
| GroupDRO | 19.5 | 98.1 |
| DANN | 21.7 | 102.9 |
| RSC | 19.5 | 98.1 |
| ANDMask | 19.5 | 98.1 |
| BCResNet | 55.3 | 154.7 |
| NSTrans | 35.3 | 75.6 |
| Koopa | 32.7 | 118.7 |
| MAPU | 46.9 | 128.3 |
| Diversify | 35.7 | 922.9 |
| Chronos | 345.5 | 1049.8 |
| Ours | 48.6 | 81.4 |

Our computation cost is comparable to the other methods, achieving much better performance. We also determine the asymptotic time complexity of the `PhASER` modules in Table 17. For multi-layer neural network modules, the representative time complexity for one layer is provided (rows 3-7).

Table 17: Complexity per module and input notation for each module.

| | Module | Complexity |
|---|---|---|
| 1 | Hilbert augmentation (using Fast-Fourier transform) | $\mathcal{O}(V \cdot N \log N)$ |
| 2 | Short-Term Fourier Transform | $\mathcal{O}(V \cdot N \cdot W \log W)$ |
| 3 | Magnitude Encoder ($F_{\mathrm{Mag}}$), Phase Encoder ($F_{\mathrm{Pha}}$), Phase Projection Head ($g_{\mathrm{Res}}$) - 2D Convolution Layers | $\mathcal{O}(k^2 \cdot N \cdot d \cdot c_{in} \cdot c_{out})$ |
| 4 | Depthwise Feature Encoder ($F_{\mathrm{Dep}}$) - 2D Convolution Layers with average pooling along feature axis | $\mathcal{O}(k^2 \cdot N \cdot d \cdot c_{in} \cdot c_{out}) + \mathcal{O}(d)$ |
| 5 | Temporal Encoder ($F_{\mathrm{Tem}}$) - (worst case backbone) Transformer Encoder | $\mathcal{O}(N \cdot d)$ |
| 6 | Classification Encoder ($g_{\mathrm{Cls}}$) - fully connected layers | $\mathcal{O}(d \cdot h)$ |

## D.7 ADDITIONAL ANALYSES

### D.7.1 TRADITIONAL AUGMENTATION

For time series, brute augmentations like scaling, reverting, cropping, and jittering may not be always suitable as they may alter the morphological properties that are important for the task. Even more advanced techniques like frequency-time warping and additive noise, need deliberate characterization of the signal's frequency response to meaningfully provide an augmented view while retaining the task-relevant semantics. This is one of the key motivating factors for us to explore a general-purpose augmentation strategy that diversifies the non-stationarity in a signal without altering its task-specific semantics (magnitude and frequency responses).

To demonstrate the use of traditional augmentations with `PhASER` for human-activity recognition, we incorporate the following augmentations proposed by past works (Qin et al., 2023; Um et al., 2017) on the HHAR dataset.

- Rotation - incorporating arbitrary rotation matrices to simulate different sensor locations.
- Permutation - random temporal perturbation for fixed window within each sample (Um et al., 2017).
- Circular Time-shift - shifting the signal by a random time interval, constrained by a predefined maximum time-shift parameter (20% of the sample length in this case) for each sample. The shifted time points from the trailing edge are wrapped around and padded to the leading edge of the signal

We incorporate these augmentations in place of the Hilbert augmentation and apply the `PhASER`. We also run an experiment with identical settings with no augmentations and illustrate in Figure 5. These results are indicative that arbitrary augmentations in the time domain do not necessarily diversify the non-stationarity of a signal. Hence, PhASER principles like residual connections to re-introduce nonstationary dictionary as phase-projection and broadcasting (using $g_{\mathrm{Res}}$) do not bode well here, and even the performance of a no-augmentation scenario is sometimes better than the traditional temporal augmentations for domain-generalization tasks in this case. However, in the future, we may encounter applications where established augmentation strategies, in combination with Hilbert augmentation, might be the best choice. In this work, we aim to propose a more generic framework that can benefit most time-series classification tasks to achieve better generalizability.

### D.7.2 RANDOM PHASE AUGMENTATION USING HILBERT TRANSFORM

We aimed to explore a random phase augmentation while ensuring minimal distortion to the signal's magnitude response to preserve important task-relevant properties. To achieve this, we leverage an adaptation of the Hilbert Transform. We illustrate our approach using a simple example: let the input signal be $\mathbf{x}(t) = \sin(\omega t)$, and its Hilbert Transform be $\mathrm{HT}(\mathbf{x}(t)) = \widehat{\mathbf{x}}(t) = -\cos(\omega t)$. For an arbitrary phase shift $\phi$, the following trigonometric identity holds:

$$\sin(\omega t + \phi) = \sin(\omega t)\cos(\phi) + \cos(\omega t)\sin(\phi). \tag{34}$$

This gives us the desired randomly phase-shifted version of $\mathbf{x}(t)$, expressed as $\mathbf{y}(t) = a\mathbf{x}(t) - b\widehat{\mathbf{x}}(t)$, where $a = \cos(\phi)$ and $b = \sin(\phi)$. The following constraint is imposed on the scalars $a$ and $b$:

$$a^2 + b^2 = 1, \tag{35}$$

which defines a valid phase shift $\phi$ as:

$$\phi = \arctan\left(\frac{b}{a}\right). \tag{36}$$

We solve for $a$ and $b$, and apply them as shown in Figure 10 to obtain an approximately identical random phase shift across all frequency components of a nonstationary signal. The desired $\phi$ is randomly sampled from the range $[-\pi/2, \pi/2]$.

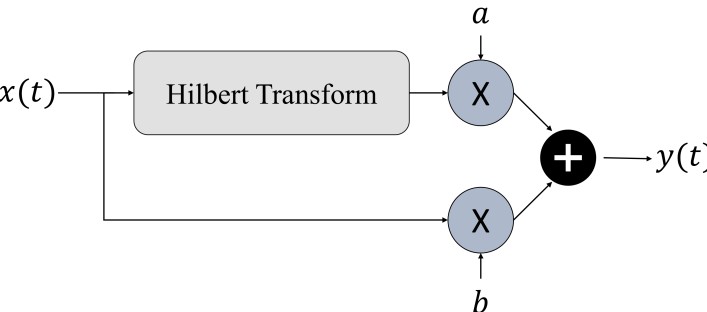

Figure 10: Schema illustrating the process for obtaining random phase augmentation by leveraging the Hilbert Transform of the original input $\mathbf{x}(t)$.

As shown in Figure 5, we observe no significant benefit from this randomization on the generalization performance of the current classification tasks. However, we are interested in exploring this direction in future by imposing additional constraints inspired by underlying processes for other time-series tasks.

### D.8 VISUALIZATION

We present some visualizations using the t-distributed stochastic neighbor embedding (t-sne) analyses on our `PhASER`, Diversify, and BCResNet for the HHAR dataset for the left-out domains in scenario 1 in Figure 6. We illustrate the t-sne plots for in-domain and out-of-domain data and the different colors indicate the six activity classes of this dataset. In all the cases, we only make necessary modifications to extract the embeddings from the last layer of the network before categorical score assignment and tune the perplexity parameters during the t-sne plotting for optimal 2-dimensional projection. Figure 11. (a,d) shows that the clustering for each class is distinct and clearly separable for both in-domain and out-of-domain data using PhASER. The accuracy disparity for unseen domains is also very low, 0.97 for in-domain PhASER accuracy and 0.94 for out-of-domain, which justifies the overall strong generalization ability of PhASER without access to any target domain samples. We would also like to point out that t-sne plots are susceptible to hyperparameters, hence, even though the accuracy of Diversify is better than BCResnet for out-of-domain data, visually Figure 11. (f) may convey better separation between classes than Figure 11. (e).

### E SUPPLEMENTARY OF MAIN RESULTS

We conduct all experiments with three random seeds (2711, 2712, 2713), and present the error range in this section. Tables 10, 11 and 12 represent the mean and standard deviation corresponding to the

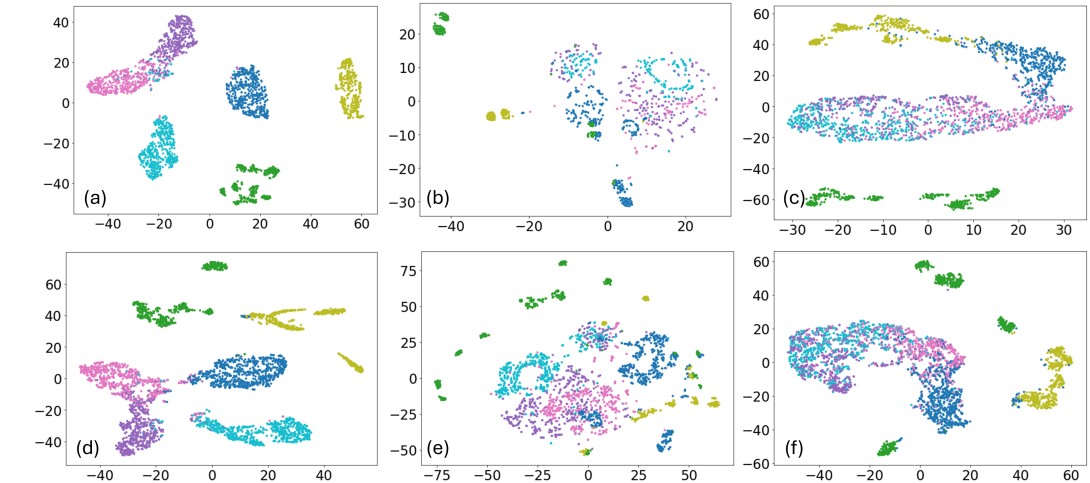

Figure 11: t-sne plots for visualizations using embeddings from HHAR scenario 1 for in-domain samples in (a) `PhASER` with an in-domain-accuracy of 0.97, (b) Diversify with in-domain accuracy of 0.82 and (c) BCResNet with in-domain accuracy of 0.78; and out-of-domain samples in (c) `PhASER` with accuracy of 0.94, (d) Diversify with accuracy of 0.77 and (e) BCResNet with accuracy of 0.74.

main paper's Table 2 for the WISDM, HHAR and UCIHAR datasets respectively. Tables 13 and 14 are the complete representations of all the runs corresponding to Table 4 in the main paper for sleep stage classification and gesture recognition respectively. Table 15 corresponds to the Table 3 in the main paper for the complete performance statistics for one person to another generalization using HHAR dataset.

# F    BROADER IMPACTS

`PhASER`, with its advanced approach to time-series domain-generalizable learning, offers significant societal benefits to various fields and domains, such as healthcare, environment monitoring, and manufacturing domains, by enabling more precise and dependable data analysis. While `PhASER` itself does not directly cause negative social impacts, its application within these critical areas necessitates a thoughtful examination of ethical concerns. In healthcare, the application of `PhASER` could usher in a new era of patient monitoring and treatment, leading to improved experiences and outcomes for individuals across diverse demographics. Its robust generalization capabilities, even with limited access to source domains (see Table 3), offer the potential to bridge gaps and foster inclusivity, particularly in minority communities, while enabling insights from rare occurrences. Moreover, for applications in environmental monitoring—ranging from continuous sensing of ambient living conditions to remote and sporadic sensing of inaccessible geological sites—`PhASER`'s principles hold promise for sample-efficient, generalizable analysis. Similarly, in manufacturing applications, `PhASER` can be deployed for both qualitative and quantitative analyses of physical components, as well as for enhancing workers' safety through continuous sensing instrumentation. However, the implementation of `PhASER` in such vital areas brings to the forefront ethical considerations like data privacy, bias prevention, and the careful management of automation reliance. Addressing these issues is important to leverage `PhASER`'s benefits across these domains while ensuring ethical integrity and maintaining public trust in these areas.

