# OpenReview forum: "Phase-Driven Domain Generalizable Learning For Nonstationary Time Series Classification"
_ICLR.cc/2025/Conference — ICLR 2025 Conference Withdrawn Submission_

### Official Review · Reviewer_QUg2 · 2024-10-25

**Soundness:** 2
**Presentation:** 1
**Contribution:** 2
**Rating:** 3
**Confidence:** 4

**Summary:**

This paper proposes a framework for the classification of non-stationary time series based on the frequency domain. The framework consists of three steps. First, the data is augmented using the Hilbert transform. Second, a feature extraction step follows, where the magnitude and phase representation of the input data are calculated based on the STFT. Finally, a neural network is used for further feature extraction and the final classification.
The key contributions of the paper are the introduction of the "PhASER" framework and extensive empirical experiments, including an ablation study.

**Strengths:**

- The combination of Hilbert transform for data augmentation, STFT for feature extraction and the proposed neural network for classification seem unique in the literature.
- Classification of non-stationary time series is a highly relevant task. The results seem promising.
- The empirical evaluation is extensive and the ablation study sheds light onto the importance of each step of the framework.

**Weaknesses:**

- The mathematical formulations are not precise; the basic definitions seem flawed.
- No theoretical guarantees of expected accuracy, true positive/negative rate etc. are provided.
- The baseline models used are not really suitable: The baseline methods are domain generalization algorithms for non-time-series-data, and the used time series models were proposed for other tasks than classification. Strong state-of-the-art classification models for time series are missing.
- Literature on non-stationary time series is abundant (also for classification). In this context, especially the generalization of stationary time series to local stationarity seems important. Further, classification methods from "functional data analysis" are relevant as well.
- The ablation study seems to suggest that using only the Hilbert transform (and omitting steps 2 and 3) yield similar results. Yet, all the differences seem minimal and could be due to randomness. No quantiles, standard deviation, or similar are reported, so it remains open how useful the steps really are.

**Questions:**

- Def. 2.1: What does the notation $Pr_{x\sim\mathcal{D}_x}(x)(t)$ mean?  The RHS of equation (1) looks like the common location-scale model. The LHS is a probability (in \[0, 1\]), are the values of the RHS expected to be in \[0, 1\] as well? What does this mean for $\mu_t, \sigma_t$ and $z$?
- Def. 2.1 excludes stationary time series. How well is the proposed method expected to work if the time series is indeed stationary?
- Def. 2.2:
	- What are $\mathcal{X}_S$ and $\mathcal{Y}_S$?
	- Are the samples independent?
	- How is determined from which source domain(s) $\mathcal{S}_j$ a sample $(x_i, y_i)$ is drawn from?
	- What is defined in "Definition 2.2"?
	- Eq. (2) suggests that each observation $(x_i, y_i)$ comes from a single source, how is $Pr(X_S, Y_S)$ related to $Pr(X_{S_i}, Y_{S_i})$?
	- $D_U$ might denote "any potential unseen target domain". Is it fixed, so the optimization is well-defined, or could it be vary?
	- The composition $F\circ g$ is interpreted as $g(F(x))$, which is not standard (usually: $(F\circ g) (x)=F(g(x))$)
- L108: "Note that the joint distributions of different source domains are similar but distinct" -> In which sense are they similar?
- L180: What is the explicit effect of the phase shift on the distribution? Of course the data looks different, but what about its distribution? (partially answered with Thm. 2.6)
- L186: Why can we assume that the time series $x$ is characterized by a deterministic function? This seems counterintuitive considering that time series are usually modeled as random
- L200: So essentially, for each time series $x(t)$, we get a second (phase-shifted) time series $\hat{x}(t)$?
- L225: What is the distribution of $p_i$? How can it be uniformly distributed on the non-negative integers (and not a finite subset)? Why should the window length $W_i$ be random at all?
- What is the overall intuition of the "magnitue-phase separate encoding"? Why is it assumed to work better than using the signal in the time domain (or the original signal concatenated with its phase shifted version)?
- Def. 2.3: What does it mean for a domain to be built on an input variable?
- Theorem 2.5: How is the risk defined?
- L343: How is it good that the distribution is changed? Is the distribution of the Hilbert-transformed output similar to the original data from some domain? If not, it seems counterintuitive that by adding "different data", the classification tasks should be simplified.
- Baseline Methods: Why are no strong baseline methods for time series classification used?
- The only reported metric is the accuracy. Are the datasets balanced, so the accuracy is a reasonable choice?
- Why is the "Related Work" section at the end of the paper? This seems to be non-standard.
- L508: "Our study is the first to rigorously address the impact of non-stationarity on time-series out-of-distribution classification" -> This is the first time that OOD classification is mentioned in the paper, how is this addressed?

---

> ### Author Response · Authors · 2024-11-16
> **Response to Reviewer QUg2 [1/2]**
>
> We appreciate your thorough feedback and recognition of the novelty of our design elements and the importance of the problem of domain generalization for nonstationary time series classification.
>
> Before addressing the more specific questions, we want to discuss our choice of baselines to address one of your key concern. Below is a breakdown of the various baselines we have carefully selected to reflect different design choices and to justify our performance in addressing this problem.
>
> * **Domain-Generalization for time-series classification**: Diversify (ICLR'23)
> * **Domain-Adaptation for time-series classification**: MAPU (KDD'23)
> * **Non-Stationary representation learning for time-series**: NSTrans (Neurips'22), Koopa (Neurips'24)
> * **Time-series Foundation Model**: Chronos (TMLR'24)
> * **Domain-Generalization for audio applications**: BCResNet (DCASE'21)
> * **Domain-Generalization for vision applications**: ERM, ANDMask, RSC, DANN, GroupDRO (standard benchmark from Gulrajani et.al in ICLR'21)
> * **PhASER + model-agnostic add-ins**: RevIN (ICLR'21)
>
> While we understand that many works can be considered relevant, it is not feasible to implement all baselines in our testbed and compare performance against every possible model. Therefore, we selected baselines that provide insights into our performance relative to specific areas of interest, such as nonstationarity modeling, domain generalization with domain labels (as in vision applications), and domain adaptation using target samples (as in MAPU). If you have a particular baseline that you believe would enhance our discussion, we would be more than willing to consider it and aim to provide timely results for your consideration during the rebuttal period.
>
> To support reproducibility and future benchmarking, we have followed the data splits established by previous works and ensured comprehensive coverage of baselines by referencing previous domain generalization work on time-series classification using these dataset splits.
>
>
> ## Questions
>
> > Definition 2.1 excludes stationary time series. How well is the proposed method expected to perform if the time series is indeed stationary?
>
> Most real-world signals with domain discrepancies are nonstationary in nature. However, if the signals were purely stationary, the task of representation learning becomes much simpler, and our method would not offer any significant advantage in such cases. For instance, consider the GR dataset, which has the smallest ADF statistics, indicating it is the least nonstationary signal among all the datasets we tested. On this dataset, PhASER and DIVERSIFY perform nearly equivalently (refer Table 4 in the main manuscript). Therefore, while our framework can be applied to all time-series domain generalization tasks, it truly shines when the signals are nonstationary.
>
> > Def. 2.2 :
>    - > What do $\mathcal{S}$ and $\mathcal{T}$ represent?
>        - The former is the marginal input distribution while the latter is the marginal label distribution.
>    - > Are the samples assumed to be independent?
>        - Yes, they are independent.
>    - > How is it determined which source domain(s) $\mathcal{S}_i$ a sample $\mathbf{x}_i$ is drawn from?
>        - It is drawn from all sources
>    - > What is being defined in "Definition 2.2"?
>        - We define the basic setups and objective of Time-Series Domain Generalization problem.
>    - > Equation (2) suggests that each observation $\mathbf{x}_i$ comes from a single source. How is this related to $\mathcal{T}$?
>        - $\mathbf{S}$ is the joint domain that unifies all $\mathcal{S}_i$.
>    - > $\mathcal{T}$ might denote "any potential unseen target domain." Is it fixed for optimization purposes, or could it vary?
>        - $\mathcal{D}_\mathrm{U}$ is not fixed, we just use it to denote a particular unseen target domain.
>    - > The composition $f \circ g$ is interpreted as $g \circ f$, which is non-standard.
>        - We put $g$ behind F to illustrate their positional relationship.
>
>
> > L108: "Note that the joint distributions of different source domains are similar but distinct." In what sense are they similar?
>
> They are similar in their spectral content but their temporal statistics are different.
>
> > L180 : What is the explicit effect of the phase shift on the distribution? While the data may appear different, how does it affect the distribution? (Partially addressed in Thm. 2.6.)
>
> The mean and variance of the distribution changes while the energy contained in time-varying frequency components is maintained.
>
> > L186 : Why is it assumed that the time series $\mathbf{x}_i$ can be characterized by a deterministic function? This seems counterintuitive, as time series are typically modeled as random processes.
>
> We claim this assumption on a time series signal observation to better define Hilbert Transform with mathematical symbols.
>
> > L200 : Does each time series $\mathbf{x}_i$ essentially produce a second, phase-shifted time series $\mathbf{x}_i'$?
>
> Yes, it does.

---

> ### Author Response · Authors · 2024-11-16
> **Response to Reviewer QUg2 [2/2]**
>
> > L225 : What is the distribution of $\mathbf{k}$? How can it be uniformly distributed on the non-negative integers rather than a finite subset? Why should the window length $w$ be random at all?
>
> $p_i$ is a random non-negative integer, and the only constraint is $2^{p_i}<\Xi$.
>
> The choice of $w$ results in varied resolution for the STFT spectra which is a hyperparameter in this case.
>
> > What is the overall intuition behind this approach? Why is it expected to work better than using the signal in the time domain, or even the original signal concatenated with its phase-shifted version?
>
> We cannot disentangle the task-irrelevant nonstationarity from the task-relevant components if we maintain only the temporal view. Translating the time series to the time-frequency spectrum allows us to leverage the magnitude response that is maintained during the augmentation and diversify the phase component and also allow it to be used as a  residual deeper in the network.
>
> > Def. 2.3 : What does it mean for a domain to be "built on an input variable"?
>
> We are referring to that a domain is defined as the data with input and label variables, $\mathbf{x}$ and $y$, respectively.
>
> > Theorem 2.5 : How is risk defined?
>
> The risk is the Gibbs risk, i.e., $R_{\mathcal{D}}[\rho]=\mathbb{E}_{(\mathbf{x}, y) \sim \mathcal{D}} \mathbb{E}_h \mathrm{I}[h(\mathbf{x})\neq y]$ where $h \sim \rho$.
>
> > L343 : Why is it beneficial that the distribution is altered? Is the distribution of the Hilbert-transformed output comparable to the original data from some domain? If not, it seems counterintuitive that adding "different data" would simplify classification tasks.
>
> The Hilbert-transformed output alters the temporal statistics while retaining the spectral information which allows the model to learn generalizable time-series representation for the task semantics.
>
> > Baseline Methods : Why are no strong baseline methods for time series classification included?
>
> We want to clarify that we are focused on domain generalization for non-stationary time-series classification and do not claim to address general time-series classification. Accordingly, our choice of datasets includes those with distinct domain discrepancies [1], and in our setting, we do not access domain labels or target samples from unseen domains. To that end, we have carefully selected datasets to reflect different design choices and to justify our performance in addressing this problem.
>
> [1] "AdaTime: A Benchmarking Suite for Domain Adaptation on Time Series Data", TKDD, 2023.
>
> > Only accuracy is reported. Are the datasets balanced, making accuracy a reasonable metric?
>
> We present the class distributions of all the datasets used in the manuscript in Figure 8 of the Appendix. Only WISDM and Sleep Stage Classification (SSC) datasets demonstrate certain imbalances among a few classes. We have presented the AUC score and the accuracy metric for the WISDM dataset in Figure 9 of the Appendix, and our conclusions remain consistent. Generally, the past works utilizing these datasets had adopted accuracy as the performance metric, and hence we adhered to the same in the submission. From our additional analyses, we can see that the choice of metric does not change our inference.
>
> > Why is the "Related Work" section positioned at the end of the paper? This structure seems non-standard.
>
> We decided to place the "Related Work" section toward the end because we wanted readers to dive into our methods and design principles right from the start. This way, the technical ideas come across more directly, with related work added afterward for broader context. Many papers have taken this approach, and it felt like the best way to tell the story here. We hope this stylistic choice doesn’t detract from the main ideas and appreciate your understanding.
>
> > L508 : "Our study is the first to rigorously address the impact of non-stationarity on time-series out-of-distribution classification." This is the first mention of OOD classification in the paper. How is this addressed?
>
> Our work focuses on domain generalization, where the model is trained on source domains using design elements such as diversification and the reintroduction of a nonstationarity dictionary, which aid in learning generalizable representations. The model is then evaluated on unseen target domains, which consist of out-of-distribution data. This nomenclature is not introduced in this paper but is instead a standard practice in works on domain generalization.

---

> > ### Comment · Reviewer_QUg2 · 2024-11-18
> >
> > Thank you for your response.
> >
> > I keep my initial rating, as I believe the key issues were not addressed: mathematical formulations are not precise, no theoretical guarantees are provided, no strong time series classification baseline model is used, crucial literature on non-stationary time series is missing, and the ablation study seems to suggest that using only the Hilbert transform yields similar results.

---

> ### Author Response · Authors · 2024-11-20
>
> We have run **a new baseline, InceptionTime** [2]—a top-ranked deep-learning time-series classification model from a recent bake-off challenge [1]—on all our datasets and included the results. It performs particularly well on one of the datasets, HHAR, and is our best baseline there. We have updated our results to reflect this appropriately. We hope this helps aid your assessment of our empirical evidence.
>
> [1] Bake-Off Redux: A Review and Experimental Evaluation of Recent Time Series Classification Algorithms, Data Mining and Knowledge Discovery, 2024.
>
> [2] InceptionTime: Finding AlexNet for Time Series Classification, Data Mining and Knowledge Discovery, 2020.
>
> To assist in your assessment, we have also added **statistics to our ablation study**.
>
> Additionally, we have made significant improvements to the mathematical notations and explanations of our theoretical justification. We hope you can refer to these updates and provide us with specific feedback on which formulations still seem imprecise to you.
>
> Your feedback is important in improving our paper for the research community, and we greatly appreciate it. Thank you once again for your time.

---

### Official Review · Reviewer_NDtH · 2024-10-28

**Soundness:** 2
**Presentation:** 2
**Contribution:** 2
**Rating:** 5
**Confidence:** 3

**Summary:**

The paper presents the PhASER (Phase-Augmented Separate Encoding and Residual) framework, designed to enhance domain-generalizable classification for nonstationary time series data. Recognizing the challenges posed by varying statistical and spectral properties in real-world applications, the authors propose a novel approach that leverages Hilbert Transform for phase augmentation, allowing for diversification of nonstationarity while preserving the discriminative semantics of the data. The framework consists of three key components: (1) phase augmentation through Hilbert Transform, (2) separate encoding of time-varying magnitude and phase responses, and (3) a broadcasting mechanism that incorporates phase information via residual connections to promote domain-invariant learning. Extensive evaluations across five datasets, including sleep-stage classification and human activity recognition, demonstrate that PhASER consistently outperforms state-of-the-art methods by an average of 5% and up to 13% in certain cases. The findings suggest that the principles of PhASER can be broadly applied to improve the generalizability of existing time-series classification models, addressing the critical need for robust pattern recognition in nonstationary environments.

**Strengths:**

This paper presents a commendable contribution to the field, demonstrating significant strengths. I believe it has two primary advantages:

1. Enhanced Generalization Across Domains: The PhASER framework effectively addresses the challenges posed by nonstationarity in time series data by leveraging phase information obtained through the Hilbert Transform. This methodology enables the model to learn domain-agnostic representations, thereby enhancing its ability to generalize across various distributions and minimizing the effects of domain shifts frequently encountered in real-world applications.

2. Robust Feature Integration: By separately encoding magnitude and phase responses, PhASER improves the integration of time-frequency information. This dual encoding strategy enables the model to more effectively capture the dynamic characteristics of time series data, resulting in enhanced classification performance. Furthermore, the implementation of a residual broadcasting mechanism reinforces the model's capability to retain critical features while mitigating the impacts of nonstationarity.

**Weaknesses:**

While this paper makes valuable contributions to the field, it also has some notable shortcomings.  I would like to highlight three primary concerns:
1. The paper employs the Hilbert Transform primarily because it effectively extracts phase information from signals and possesses non-parametric properties, making it particularly useful for handling nonstationary time series.  However, wavelet transform offers significant advantages in processing nonstationary signals, especially in time-frequency analysis and multi-resolution feature extraction.  Therefore, why is wavelet transform not utilized in this context?
2. The paper highlights the importance of phase information in the classification of nonstationary time series.  Could you elaborate on how phase information influences the model's learning process?  What specific roles does phase information play in different application scenarios?
3. Image augmentation techniques, such as rotation, may significantly impact the input frequency and phase modalities, particularly when processing time series data.  How can I determine the extent of these transformations to enhance performance without compromising the characteristics of the original data?

**Questions:**

The authors present their ideas with remarkable clarity; however, I have one question: Could the authors provide a more detailed explanation of the benefits of the Phase-driven Residual Broadcasting method employed in this study, particularly in comparison to other multimodal fusion approaches? What specific advantages does it offer in this context?

---

> ### Author Response · Authors · 2024-11-15
> **Response to Reviewer NDtH [1/2]**
>
> We thank you for your time, and feedback and for commending our work on originality and contribution. We would like to respond to your questions and concerns below.
>
> ## Weaknesses:
>
> > The paper employs the Hilbert Transform primarily because it effectively extracts phase information from signals and possesses non-parametric properties, making it particularly useful for handling nonstationary time series. However, wavelet transform offers significant advantages in processing nonstationary signals, especially in time-frequency analysis and multi-resolution feature extraction. Therefore, why is wavelet transform not utilized in this context?
>
> We want to walk you through our design intuitions and justify our choice of using the Hilbert Transform's analytic expression to conduct a phase shift in nonstationary signals:
>
> * Wavelet transform can help in applying random phase shifts at localized temporal scales, but the choice of the mother wavelet which is application-specific heavily governs the performance of this transform. In contrast, the Hilbert transform is very simple and ensures consistent magnitude response too, making it an attractive choice across various applications for conducting phase augmentation.
>
> * As for the randomness of the phase augmentation within a signal, our preliminary analysis in Section 3.2, Figure 5, and the design explained in Section D.7.2 in the Appendix (Figure 10) using a modified version with Hilbert transform was not particularly beneficial. So, we don't find the motivation to go to extra lengths to discover a generally good wavelet to facilitate random phase augmentations through Wavelet transform.
>
> * Although, we did explore the empirical model decomposition technique where we could extract instantaneous frequency, and by combining it with the complex representation of the signal, we can achieve instantaneous amplitude and phase thus facilitating any arbitrary phase shifts too. However, our initial results were not promising as they heavily depended on the choice of the number of intrinsic mode functions based on the application. We briefly discuss this in Section B of the Appendix (lines 1073-1078).
>
> But, we agree that a wavelet transform is an interesting tool that can be harnessed to explore other time-frequency representations instead of the STFT if we want to employ special signal processing elements for a given application, such as multirate signal analysis or localization of a particular frequency component. However, for purposes of a generic way to conduct phase-diversification through in-sample augmentation, we advocate the use of Hilbert Transform.
>
>
> > The paper highlights the importance of phase information in the classification of nonstationary time series. Could you elaborate on how phase information influences the model's learning process? What specific roles does phase information play in different application scenarios?
>
> Our intuition is that phase can serve as an approximate proxy for non-task-specific nonstationarities. We refer to Table 1, row 2, where using only phase features results in the lowest accuracy for activity recognition in the WISDM dataset. Task-specific nonstationarities are characterized through the Short-Time Fourier Transform (STFT), which ensures that information is not lost and allows the neural network to learn it if it aids in optimizing the objective. We demonstrate this intuition through our design process and show empirical evidence on 5 datasets and 11 baselines.
>
>
>
> > Image augmentation techniques, such as rotation, may significantly impact the input frequency and phase modalities, particularly when processing time series data. How can I determine the extent of these transformations to enhance performance without compromising the characteristics of the original data?
>
>
> Yes, the direct application of augmentation techniques from the vision domain may not be suitable for all the time-series applications. Particularly for rotation, some past works on Human-activity recognition (HAR) [1] including a very recent work [2], suggest that it can help emulate different positions of placement of the Inertial-Measurement-Unit (IMU) which is usually the key sensor for wearable HAR. Therefore, we also explore it in the context of HHAR dataset briefly in Figure 5.
>
>
> We do not attempt to propose a new type of time-series augmentation for general time-series representation learning; rather, we present our findings that phase anchoring through multiple stages supports domain generalization in time-series classification tasks, with augmentation being one of those stages.
>
> [1] Generalizable low-resource activity recognition with diverse and discriminative representation learning, KDD, 2023
>
> [2] UniMTS: Unified Pre-training for Motion Time Series, Neurips, 2024

---

> > ### Comment · Reviewer_NDtH · 2024-11-16
> >
> > Thank you for your response.
> >
> > Regarding the second question, I still have some doubts: the authors mentioned, "Our intuition is that phase can serve as an approximate proxy for non-task-specific nonstationarities." I would appreciate further clarification on how this intuition was developed. In other words, as Reviewer VbRq described, many characteristics of the signal may exhibit changes in "non-stationary statistics." Why was phase information specifically chosen to capture these changes?

---

> ### Author Response · Authors · 2024-11-15
> **Response to Reviewer NDtH [2/2]**
>
> ## Questions:
>
> > The authors present their ideas with remarkable clarity; however, I have one question: Could the authors provide a more detailed explanation of the benefits of the Phase-driven Residual Broadcasting method employed in this study, particularly in comparison to other multimodal fusion approaches? What specific advantages does it offer in this context?
>
> A residual broadcasting-style network is used in many audio-event detection applications and is known to support generalizable learning while remaining conservative in model size [4, 5]. It may not be accurate to compare it with general multimodal fusion architectures (early fusion, late fusion, etc.), as we do not have distinct modalities but rather distinct representations (magnitude and phase) of the same modalities in the residual-broadcasting connection. It is more comparable to residual connections since this broadcasting unit can be repeated several times if the network is deeper.
>
>
> [3] Broadcasted Residual Learning for Efficient Keyword Spotting, Interspeech, 2021
>
> [4] Domain Generalization on Efficient Acoustic Scene Classification Using Residual Normalization, DCASE, 2021

---

> ### Author Response · Authors · 2024-11-16
>
> Thank you for the question.
>
> We do not mean to suggest that phase is the only way to assess the non-stationarity of a signal (as noted in our response to Reviewer VbRq). Through this work, we aim to demonstrate that phase can be diversified in a way that does not alter the spectral properties of a signal and by incorporating this diversified phase-derived embedding deeper into the network to emulate more "difficult" non-stationarities possible from unseen target domains, we can ensure that the model learns only task-relevant semantics overcoming the irrelevant nonstationarities for time-series classification. Our overall design intuition is supported by the controlled pilot study in Figure 3, where, despite increasing the non-stationarity of a signal, PhASER can perform consistently well.

---

> > ### Comment · Reviewer_NDtH · 2024-11-17
> >
> > Thank you for your patience to response. Unfortunately, based on the content of the article, I must maintain my rating. I encourage the authors to continue their efforts and consider submitting this excellent work to a more suitable journal or conference.

---

> > > ### Author Response · Authors · 2024-11-17
> > >
> > > Thank you for your suggestion, but we noticed a change in your original assessment of our work and would appreciate understanding the specific reasons behind the decrease in score. This information is crucial for us to address any issues and improve our work. We are committed to making the necessary revisions and would value your guidance on how we might restore the original, higher assessment.

---

> > > > ### Comment · Reviewer_NDtH · 2024-11-17
> > > >
> > > > I have always considered your work to be highly promising. However, after reviewing the feedback from other experts, I identified an issue that was overlooked during my initial review: the use of phase information to capture changes in "non-stationary statistics." As a result, I felt the need to adjust my initial assessment. Although you provided a response to this concern, I still find the rationale for specifically choosing phase information to capture these changes insufficiently explained. The underlying reasoning behind this choice remains unclear, which led me to revise my evaluation.

---

> > > > > ### Author Response · Authors · 2024-11-17
> > > > >
> > > > > We appreciate your response.
> > > > >
> > > > > We want to reiterate that phase is not the only way to update the nonstationary statistics, but it is one of the potential methods that we are showcasing in this work. There have been no prior works that have explored this perspective, and we aim to demonstrate a simple phase-anchored design in three stages to handle nonstationary time-series in domain generalization for classification tasks: 1) data augmentation, 2) phase-magnitude separate encoding, and 3) phase broadcasting.
> > > > >
> > > > > We urge you to also check our responses to other reviewers' comments while we wait for them to engage with us. We believe we have resolved their primary concerns. In any case, we thank you for your time.

---

### Official Review · Reviewer_VbRq · 2024-11-03

**Soundness:** 2
**Presentation:** 2
**Contribution:** 2
**Rating:** 3
**Confidence:** 3

**Summary:**

The paper presents a new network for nonstationary time series classification with domain generalization capabilities. The new network is based on a separate processing of the phase of the signal, which is extracted using the HT and employed in parallel to the processing of the magnitude of the signal. The advantages of the approach are demonstrated through experiments on several benchmarks.

**Strengths:**

- To the best of my knowledge, incorporating phase information in this manner for the purpose of time-series classification is new.
- The empirical evaluation is extensive, showing improvements over other methods across several benchmarks from different application domains.

**Weaknesses:**

- The proposed approach, which adds phase information using the HT that in turn is processed in parallel with the signal's magnitude, is relatively simple. On its own, this may not be sufficient to justify publication in ICLR.
- The theoretical justification in Section 2.5 is unclear, and Theorem 2.6, in particular, is unconvincing. Numerous features of a signal, other than the phase, could exhibit changes in "nonstationary statistics”. Additionally, the term "nonstationary statistics" is used frequently throughout the paper but is not clearly defined.

**Questions:**

- I would like to ask the authors to clarify the meaning of their theoretical results, specifically a clearer explanation of Theorem 2.6 and a formal definition of "nonstationary statistics”.

---

> ### Author Response · Authors · 2024-11-15
> **Response to Reviewer VbRq**
>
> We appreciate your recognition of the novelty in our approach and thank you for your time and feedback on improving our work. We address your concerns below.
>
> > The proposed approach, which adds phase information using the HT that in turn is processed in parallel with the signal's magnitude, is relatively simple. On its own, this may not be sufficient to justify publication in ICLR.
>
> While we propose a simple, in-sample, non-parametric augmentation using the Hilbert Transform, our work also introduces two key model-related designs: separate magnitude-phase feature encoding and the use of phase residuals in a broadcasting operation. We evaluate our approach comprehensively across five datasets and against 11 diverse baselines, some of which are more complex, assume access to target domain samples, or leverage domain labels in their learning. Despite these advantages for the baselines, our method outperforms them in generalization performance.
>
> As noted by you, "incorporating phase information in this manner for the purpose of time-series classification is new." Through this work, we aim to showcase the value of integrating signal-processing concepts into time-series deep learning analyses and present our findings on the advantage of a phase-anchored design scheme for generalizable representation learning in time-series classification tasks without accessing target samples or domain labels. Additionally, we provide benchmarks (leveraging data splits proposed by past works [1] and re-running older baselines on them) that the community can use for future research in this direction.
>
> We believe that simplicity is a key strength of our approach, and we systematically demonstrate the value of each (carefully designed) component through ablation studies, supported by theoretical grounding. We hope that the community finds these ideas beneficial.
>
>
> [1] "AdaTime: A Benchmarking Suite for Domain Adaptation on Time Series Data", TKDD, 2023.
>
> > The theoretical justification in Section 2.5 is unclear, and Theorem 2.6, in particular, is unconvincing. Numerous features of a signal, other than the phase, could exhibit changes in "nonstationary statistics”. Additionally, the term "nonstationary statistics" is used frequently throughout the paper but is not clearly defined.
>  > Question : I would like to ask the authors to clarify the meaning of their theoretical results, specifically a clearer explanation of Theorem 2.6 and a formal definition of "nonstationary statistics”.
>
> We would like to draw your attention to **Definition 2.1, especially Equation 1**, where we define nonstationary time series. Regarding Theorem 2.6, we agree that attributes other than phase can contribute to a signal's nonstationarity. Therefore, we do not claim that phase is the only way to diversify the source domains' nonstationarity, but rather one of the ways.
>
> We would also like to reiterate that our design takes into account the spectral nonstationarity inherent in real-world signals. While formally proving this is challenging without conducting signal decomposition under system-specific assumptions—unlike statistical nonstationarity, which can be characterized for representative samples using Augmented Dickey-Fuller (ADF) tests, as shown in Table 8 of the Appendix—our approach leverages transforms such as STFT and HT. These are standard tools for analyzing nonstationary or multirate signals while preserving the energy content of time-varying frequency components that are likely relevant to task semantics. By modifying only the phase, our method introduces diversity in temporal nonstationarity, thereby enriching the representation space without compromising the spectral characteristics critical for nonstationary time series classification.
>
> We would greatly appreciate your further specific feedback on this justification to help improve our paper.

---

> > ### Author Response · Authors · 2024-11-20
> > **Gentle Reminder for Feedback on our Response**
> >
> > As we are in the middle of the response period, we wanted to reach out to see if your concerns have been resolved or if you have any additional questions or remarks for us.
> >
> > Please consider our revised manuscript in your further assessment.

---

> > > ### Comment · Reviewer_VbRq · 2024-11-25
> > >
> > > I would like to thank the authors for the detailed responses and their engagement.
> > > However, my main concern remains. I believe a stronger theoretical explanation is needed to clarify the incorporation of phase information in the proposed manner.

---

### Official Review · Reviewer_WYhL · 2024-11-04

**Soundness:** 1
**Presentation:** 2
**Contribution:** 2
**Rating:** 3
**Confidence:** 3

**Summary:**

This article describes a data augmentation procedure to create new time series using the magnitude and phase information of the short-term Fourier transform (STFT). The authors also propose to separately encode the magnitude and phase, and specific architecture ("phase residual feature broadcasting") to perform downstream tasks (such as classification).

**Strengths:**

The study of non-stationary time series is an important topic in real applications. The proposed approach is an interesting addition to the augmentation strategies that create new time series with similar dynamics.

**Weaknesses:**

There are many inaccuracies in the manuscript, making it difficult to follow.
- Many definitions and statements are not mathematically sound.
  * Equation 1: Left-hand size is a function, and right-hand size is a random variable.
  * Definition of HT: It should be HT(x)(t) and not HT(x(t)) unless the authors mean that HT operates on a real value.
  * $\mathbf{x}= \\{ x_0,\dots,x_t,\dots \\} \in\mathbb{R}$
  * Definition of $\mathcal{D}_{\bar{U}}$ in Theorem 2.5
- When doing the ADF test, the authors should provide the p-values instead of the raw statistics, which are not interpretable.
- In Definition 2.2, is there a connection between $S_{i} $ and $\mathcal{X}_{{S}}$?
- The definition of non-stationarity differs from what can be found in the literature. Usually, the definition of (2nd order or weak) stationarity is given, and non-stationary time series are time series that do not satisfy this definition.
- The authors introduce a beta divergence, which is not the usual one [1]. Furthermore, it does not satisfy the properties of a divergence. Also, Equation 9, on which the theoretical analysis relies, must be clearly stated (not all terms are defined).
- There needs to be more clarity between the claims and what the proposed methodology can achieve.
  * The authors claim that PHASER is designed for domain adaptation, but which proposed mechanism contributes to domain adaptation?
  * The augmentation strategy takes a non-stationary time series and returns another non-stationary time series. None of the methodology's other components are designed for non-stationary time series. The authors should be more specific about why the whole pipeline is adapted for non-stationary time series.
  * The feature broadcasting mechanism is described purely from an implementation standpoint. There should be a more intuitive explanation of what it does, with some evidence or relation to the literature.
- There is extensive literature on time series classification [2]. The choice of baselines needs to be more motivated as standard methods were excluded. Also, close to half of the baselines are from non-published articles.
- There are poorly formulated sentences which can be surprising for readers ("the joint distributions […] are similar but distinct.", "diversify non-stationarity," "DFT is applicable for signals that are stationary and periodic."


[1] Hennequin, R., David, B., & Badeau, R. (2011). Beta-divergence as a subclass of Bregman divergence. IEEE Signal Processing Letters, 18(2).

[2] Ruiz, A. P., Flynn, M., Large, J., Middlehurst, M., & Bagnall, A. (2021). The great multivariate time series classification bake off: a review and experimental evaluation of recent algorithmic advances. Data Mining and Knowledge Discovery, 35(2).

**Questions:**

See my comments in the section above.

---

> ### Author Response · Authors · 2024-11-16
> **Response to Reviewer WYhL [1/2]**
>
> We thank you for your time and feedback. We address your main concerns below.
>
> >Many definitions and statements are not mathematically sound.
>
> In Eq.(1), the left side represents the probability density function of the variable on the right side.
>
>
> We have assumed the real-valued signal is characterized by a function $x(t)=\mathbf{x}_t$, thus use $x(t)$ in the definition of HT.
>
>
> > When doing the ADF test, the authors should provide the p-values instead of the raw statistics, which are not interpretable.
>
> We have updated Table 8 in the Appendix to include p-values. Please refer to the updated pdf. Initially, we only reported the ADF statistics, following the reporting standards used in prior works [1, 2]  on nonstationarity.
>
> [1] "Non-stationary transformers: Exploring the stationarity in time series forecasting", Neurips, 2022.
>
> [2] "Koopa: Learning Non-stationary Time Series Dynamics with Koopman Predictors", Neurips, 2024.
>
> > In Definition 2.2, is there a connection between $\mathcal{S}_i$ and $\mathcal{X}_\mathbf{S}$?
>
> Yes, there is, as $\mathcal{X}_\mathbf{S}$ is the marginal input distribution of the joint domain that unifies all $\mathcal{S}_i$.
>
> > The authors introduce a beta divergence, which is not the usual one. Furthermore, it does not satisfy the properties of a divergence. Also, Equation 9, on which the theoretical analysis relies, must be clearly stated (not all terms are defined).
>
> The widely used divergence in cross-domain learning is defined on hypothesis disagreement (based on task module)[3], which cannot characterize the nonstationarity in the raw features of time series. Beta-divergence is defined on the raw data space, thus suitable for our problem. All terms in Eq. (9) have been defined, $q$ is a non-negative constant while $\mathrm{RD}(\cdot)$ is the Renyi divergence.
>
> [3] Mansour, Y., Mohri, M., and Rostamizadeh, A. Domain adaptation: Learning bounds and algorithms. In COLT, 2009.
>
> > There needs to be more clarity between the claims and what the proposed methodology can achieve.
>
> We want to clarify that we are doing domain generalization (without accessing the domain labels) and not domain adaption which assumes access to target domain labels.
>
> Our simplified intuition is as follows: after phase diversification while retaining the magnitude spectrum, we perform separate magnitude-phase encoding. Subsequently, we reintroduce the nonstationarity dictionary (diversified phase-derived embeddings) as a residual connection deeper in the network. This step mimics the aggressive task-irrelevant nonstationarity of unseen domains by corrupting the learned embeddings with phase embeddings—representations that are only weakly relevant to the task. If our model can still learn meaningful representations, it will generalize well.
>
> Regarding more justification and literature for residual broadcasting-style network, it is often used for on-device audio-event detection applications and is known to support generalizable learning while maintaining a conservative model size [4, 5]. Design elements like sub-spectral normalization [6] help disambiguate high- and low-frequency information, enabling device-related generalization to account for factors such as sensor drifts or other slowly varying components (like Direct Current, DC) that typically manifest in the lower frequency spectra. Inspired by this intuition, applied in audio domains with Mel-Frequency Cepstral Coefficients (MFCCs), we extend this approach to multivariate time-series data using an STFT-based time-frequency representation, incorporating a phase-based residual with a more phase-anchored design.
>
> [4] Broadcasted Residual Learning for Efficient Keyword Spotting, Interspeech, 2021
> [5] Domain Generalization on Efficient Acoustic Scene Classification Using Residual Normalization, DCASE, 2021
> [6] SubSpectral Normalization for Neural Audio Data Processing, ICASSP, 2021

---

> > ### Author Response · Authors · 2024-11-16
> > **Response to Reviewer WYhL [2/2]**
> >
> > > There is extensive literature on time series classification. The choice of baselines needs to be more motivated as standard methods were excluded. Also, close to half of the baselines are from non-published articles.
> >
> > We want to clarify that we are focused on domain generalization for non-stationary time-series classification and do not claim to address general time-series classification. Accordingly, our choice of datasets includes those with distinct domain discrepancies [7], and in our setting, we do not access domain labels or target samples from unseen domains. To that end, we have carefully selected datasets to reflect different design choices and to justify our performance in addressing this problem.
> >
> > **None** of the baselines we have chosen are unpublished. Below is a breakdown of the baselines and the venues where they were presented.
> >
> > * **Domain-Generalization for time-series classification**: Diversify (ICLR'23)
> > * **Domain-Adaptation for time-series classification**: MAPU (KDD'23)
> > * **Non-Stationary representation learning for time-series**: NSTrans (Neurips'22), Koopa (Neurips'24)
> > * **Time-series Foundation Model**: Chronos (TMLR'24)
> > * **Domain-Generalization for audio applications**: BCResNet (DCASE'21)
> > * **Domain-Generalization for vision applications**: ERM, ANDMask, RSC, DANN, GroupDRO (standard benchmark from Gulrajani et.al in ICLR'21)
> > * **PhASER + model-agnostic add-ins**: RevIN (ICLR'21)
> >
> > [7] "AdaTime: A Benchmarking Suite for Domain Adaptation on Time Series Data", TKDD, 2023.

---

> > > ### Comment · Reviewer_WYhL · 2024-11-18
> > >
> > > `None of the baselines we have chosen are unpublished. Below is a breakdown of the baselines and the venues where they were presented.`
> > >
> > > Thanks for the clarification. In that case, the references should be updated. In the current version, only the ArXiv information is provided.

---

> > ### Comment · Reviewer_WYhL · 2024-11-18
> >
> > Thanks for the clarifications, and I appreciate the authors' effort in responding to my comments. This helped me understand a couple of notations, although they still need to be corrected in the paper. Rigorous notations will help readers focus on the actual contribution.
> > For instance, the notation in Eq. 1 ($P(\mathbf{x}) = \mu + \sigma z$) is still ambiguous. What would $P(\mathbf{x}) + P(\mathbf{x}^\prime)$ represent: the sum of two probabilities or the sum of two vectors? Also, $z$ is said to be a "stationary stochastic component." If it is a stationary process, it should be indexed by a time variable.
> >
> > Similarly, the $\beta$-divergence is a well-known function, different from the one introduced here. A comment explaining why the authors modified this classical notion would greatly help readers.
> >
> > I have additional questions/remarks:
> > - In Theorem 2.5, $\mu_{i,t}$ and $\sigma_{j,t}$ are not defined.
> > - In the following paragraph ("Insights"), the authors connect Theorem 2.5 to the choice of adding "a phase-residual connection." However, this needs to be clarified for me. In particular, the following sentence should be more precise: "to associate minimization of $\beta$-Divergence with the classification task via a phase-residual connection that is an approximate proxy of domain-specific non-stationarity."
> > - The Hilbert transform is defined everywhere for continuous time series ($t\in\mathbb{R}$), but we observe discrete time series ($t\in\mathbb{N}$). Does any of the theoretical results hold over the discrete version?
> > - Theorem 2.6 implies that the distribution of the augmented time series is different from the original time series. Can the authors clarify why this is important for the task at hand?
> >
> > To improve my rating, I will wait for a significant improvement in the general presentation of the mathematical content and a more precise theoretical justification of why the proposed method tackles non-stationarity and domain shifts.

---

> ### Author Response · Authors · 2024-11-20
>
> We thank you for your response and patience while we address your concerns.
>
> 1. $\mu_{i, t}$ and $\sigma_{i,t}$ are defined in Definition 2.1 and represent the mean and variance of time-series signals from domain $\mathcal{S}_i$ at timestamp , $t$, respectively.
>
> We have also highlighted other definitions, such as $\mathcal{D}_\mathrm{U}$, which were pointed out as undefined but are already described in Definition 2.2. We have also added an explanation to lines 108-109, and 160-161 upon your advice for clarity. Please let us know if you have any other notation-specific concerns, and we will address them promptly.
>
>
> 2. Regarding further theoretical justification about Theorem 2.5 : As mentioned in the paper, we regard $\beta$-divergence as a proxy of non-stationarity, but it is infeasible to approximate it in the raw feature space. Therefore we approximate the $\beta$-divergence in the representation space, and specifically, we approximate it in the low-level representation space extracted by the phase feature encoder $F_{\mathrm{Pha}}$ and in the high-level representation space extracted by the temporal feature encoder $F_\mathrm{Tem}$. To better minimize these two levels of approximations, we connect them via a residual connection. We have clarified this in the updated manuscript.
>
> 3. Regarding further explanation of Theorem 2.6: By using phase augmentation to modify the input data statistics, we encourage the model to learn task-relevant semantics despite diverse conditions of non-stationarity, which helps in learning more generalizable representations, ultimately leading to higher performance even for unseen target domains.
>
> 4. We have updated all the references to reflect their final venue of publication (except Chronos which got accepted to TMLR on 11 November and does not have a Digital Object Identifier yet).
>
> 5. Additionally, we have run a new baseline, InceptionTime [2] -- top-ranked deep-learning time-series classification model from a recent bake-off challenge [1] -- for all our datasets and included the results. We hope this helps in aiding your assessment of our empirical evidence.
>
> [1] Bake off redux: a review and experimental evaluation of recent time series classification algorithms, Data Mining and Knowledge Discovery, 2024
>
> [2] InceptionTime: Finding AlexNet for Time Series Classification, Data Mining and Knowledge Discovery, 2020

---

> > ### Author Response · Authors · 2024-11-23
> >
> > Dear Reviewer,
> >
> > As we approach the end of the response period, we wanted to follow up to see if your concerns have been addressed or if you have any additional questions.
> >
> > We kindly request that you consider our revised manuscript in your further assessment.

---

> > > ### Comment · Reviewer_WYhL · 2024-11-25
> > >
> > > I thank the authors for the answers and appreciate their effort. The theoretical reasons why the proposed approach can handle non-stationarity and domain shift remain unclear in the revised manuscript, so I will maintain my rating.

---

### Official Review · Reviewer_TvsP · 2024-11-04

**Soundness:** 2
**Presentation:** 2
**Contribution:** 2
**Rating:** 5
**Confidence:** 2

**Summary:**

This paper proposes a phase augmentation method to improve the robustness of the time-series prediction. The method is based on the Hilbert Transform, which does not change the magnitude response. Therefore, it is claimed that this augmentation can diversify the dataset while preserving discriminative features. Experiments are conducted on several datasets.

**Strengths:**

The domain generalization for time-series data is interesting. The experimental results verify the effectiveness of the proposed method. The organization is clear, although some details are not introduced clearly.

**Weaknesses:**

My primary concern/question is, why changing the phase would preserve discriminative features. The implicit assumption is that the response is mainly determined by its magnitude. But without introducing how y is generated, it looks confusing to say that phase augmentation preserves discriminative features.

Besides, the model assumes piecewise constant distribution in Eq. (1). Is it able to relax this assumption? It seems that many previous methods do not assume this, e.g., Diversify.

Third, the theoretical part seems redundant. Its connection to the method is weak. Removing this part also makes the method self-motivated.

**Questions:**

Have you compared your method with the Diversify method in the augmentation step?

---

> ### Author Response · Authors · 2024-11-15
> **Response to Reviewer TvsP [1/2]**
>
> We thank you for taking the time to read and provide feedback on our paper. However, we would like to clarify a few points that may have been misunderstood:
>
> 1. Our paper does not focus on introducing a new type of time-series augmentation for general time-series representation learning; rather, it presents our findings that phase anchoring through multiple stages supports domain generalization in time series classification tasks, with augmentation being one of those stages.
>
> 2. While we propose a simple in-sample, non-parametric augmentation using the Hilbert Transform, we also introduce two other model-related designs: separate magnitude-phase feature encoding and the use of phase residuals for a broadcasting operation.
>
> 2. Beyond a comprehensive evaluation across 5 datasets and against 11 baselines, we systematically demonstrate the value of each component through ablation studies. We reiterate the respective average improvements achieved by separate phase-magnitude encoding and phase residuals, as shown in Table 5 below.
>
>
> | **Residual Connection Version**                                    | **Average Improvement (%)** |
> |--------------------------------------------------------------------|-----------------------------|
> | Only Magnitude Residual (Row 3)                                   | 9%                          |
> | Only Magnitude Input with Magnitude Residual (Row 7)              | 4.5%                        |
> | Phase-Magnitude Fusion Residual (Row 4)                            | 5.5%                        |
> | No Residual Connection (Row 5)                                    | 4%                          |

---

> > ### Author Response · Authors · 2024-11-15
> > **Response to Reviewer TvsP [2/2]**
> >
> > Below we address your concerns and questions.
> >
> > ## Weaknesses
> > > My primary concern/question is, why changing the phase would preserve discriminative features. The implicit assumption is that the response is mainly determined by its magnitude. But without introducing how y is generated, it looks confusing to say that phase augmentation preserves discriminative features.
> >
> > We do not assume that only magnitude contains discriminatory characteristics. In fact, the evidence in Table 1, where phase-only features achieve an accuracy of 0.62 in a six-class classification task—significantly higher than chance accuracy—supports the presence of task-discriminating attributes in the phase response. However, this accuracy is notably lower than that achieved with magnitude-only features (0.82). Our augmentation strategy, which uses the analytic signal from the Hilbert Transform, has the property of preserving the magnitude response for any arbitrary nonstationary signal. Hence, we say that phase augmentation can preserve task-relevant semantics for time series classification tasks.
> >
> >
> > > Besides, the model assumes piecewise constant distribution in Eq. (1). Is it able to relax this assumption? It seems that many previous methods do not assume this, e.g., Diversify.
> >
> > A piecewise constant distribution is a reasonable assumption for approximating most real-world sensing signals. The step where this assumption manifests during implementation is the choice of window size for the STFT of the signal, which can be adjusted according to application-driven specifications.
> >
> > Previous works, such as Diversify [1] and AdaRNN [2], which address domain generalization from a distributional perspective, attempt to uncover latent distributions by identifying sub-segments within a segment where distributions are constant. However, they rely on an assumption or a chosen hyperparameter regarding how many such latent distributions are present in a segment.
> >
> > Our approach, PhASER, is anchored in the phase response of a signal, following our intuition that it captures a dictionary of the signal's nonstationarity. By diversifying, separately encoding, and reintroducing the phase latent as a residual in a broadcasting operation, we are able to learn robust, generalizable time-series representations for classification.
> >
> > Another supporting argument for our approach is that by not explicitly characterizing nonstationarity but rather diversifying its content (through phase augmentation) and compelling the model to perform the given task despite the reintroduction of nonstationarity-related features deeper in the layers (through phase-residual connections), we enhance generalization performance by minimizing the $\epsilon$, which is influenced by the nonstationary statistics as shown in Eq. (13). This, in turn, minimizes the overall risk, $R_{\mathcal{D}_\mathrm{U}}$, as shown in Eq. (11).
> >
> >
> > [1] "Out-of-distribution representation learning for time series classification", ICLR, 2023.
> >
> > [2] "AdaRNN: Adaptive Learning and Forecasting for Time Series", Proceedings of the 30th ACM international conference on information & knowledge management,2021.
> > > Third, the theoretical part seems redundant. Its connection to the method is weak. Removing this part also makes the method self-motivated.
> >
> > The theoretical justification grounds our empirical evidence with established theory, offering readers a well-rounded perspective on our method. We have intentionally structured the manuscript to make PhASER practically accessible without requiring readers to engage deeply with the theoretical foundations.
> >
> >
> > ## Questions
> >
> > > Have you compared your method with the Diversify method in the augmentation step?
> >
> > We aim to maintain the paper's focus on Domain Generalization for Time-Series Classification, rather than shifting it toward proposing a new augmentation for general time-series learning (which is our follow-up work). Therefore, we do not conduct an extensive evaluation focused solely on augmentation but instead provide a pilot analysis with other common strategies in Section 3.2, Figure 5, to aid the reader's understanding of our method. Overall, we include Diversify as a baseline (finding it to be one of the strongest baselines across 4 out of 5 datasets) in all our evaluations.

---

> > > ### Author Response · Authors · 2024-11-20
> > > **Gentle Reminder for Feedback on our Response**
> > >
> > > As we are in the middle of the response period, we wanted to reach out to see if your concerns have been resolved or if you have any additional questions or remarks for us.
> > >
> > > Please consider our revised manuscript in your further assessment.

---

> > > > ### Comment · Reviewer_TvsP · 2024-11-22
> > > >
> > > > Thank you for your response. However, most of my concerns have not been addressed. For example, it is still not convincing why changing the phase would preserve discriminative features. Besides, the piecewise constant assumption restricts the scope of applications.

---

> > > > > ### Author Response · Authors · 2024-11-22
> > > > >
> > > > > We thank you for your response.
> > > > >
> > > > > We want to add a further comment justifying the role of phase augmentation.
> > > > >
> > > > > Only the considering the phase of a signal for classification tasks endures heavy information loss as evident from its low performance of 0.62 in comparison to magnitude 0.82. So, our motivation is that augmenting phase does not heavily corrupt the signal's task-specific semantic properties. Past works also support such an approach where phase is used in mixup strategy [1].
> > > > >
> > > > > [1] Finding Order in Chaos: A Novel Data Augmentation Method for Time Series in Contrastive Learning, Neurips 2024.

---

### Author Response · Authors · 2024-11-16
**Global Response to Reviewers**

We sincerely thank all reviewers for their time and thoughtful feedback.

In response to Reviewer WYhL's comments, we have included the p-values for the ADF test in Table 8 of the Appendix and clarified the rationale behind our choice of baselines and their publication venues, addressing the concern about using unpublished baselines, which is not the case.

We welcome additional feedback, particularly from Reviewers QUg2 and WYhL, regarding the mathematical notations used. We hope our responses have addressed your questions clearly. In any case, such notation-related changes/further justifications can be readily incorporated during the rebuttal phase and do not detract from our core message: that the phase-anchored design has the potential to deliver a generalizable time-series representation for classification tasks.

If there are areas where our responses could be improved or specific points that warrant further clarification to merit a higher score, we would greatly appreciate your insights. Your feedback is valuable in refining our work.

Even if you maintain your initial assessments, we value the opportunity to better understand your concerns and look forward to a constructive discussion.

Thank you once again for your review!

---

### Note · Authors · 2025-01-13

I have read and agree with the venue's withdrawal policy on behalf of myself and my co-authors.